# Plants with higher dispersal capabilities follow 'abundant-centre' distributions but such patterns remain rare in animals

Connor T. Panter [1] ✉, Stephan Kambach [2], Steven P. Bachman [3], Oliver Baines[1,4], Helge Bruelheide [2,5], Maria Sporbert[2], Georg J. A. Hähn [6], Richard Field[1] & Franziska Schrodt [1]

The 'abundant-centre' hypothesis posits that a species' abundance is highest at its range centre and declines towards its range edge. Recently, the hypothesis has been much debated, with supporting empirical evidence remaining limited. Here, we test the hypothesis on 3660 species using 5,703,589 abundance observations. We summarise species-level patterns and test the effects of dispersal-related species traits and phylogeny on abundance–distance relationships. Support for the hypothesis is dependent on taxonomic group, with abundant-centre patterns being more pronounced for plants but non-significant when summarised across all animals. Dispersal capability does not explain abundance–distance relationships in animals but likely explains abundance patterns in non-woody plants. Phylogeny improves models of abundance–distance patterns for plants but not for animals. Despite this, controlling for phylogeny yields non-significant group-level results for plants, suggesting that only certain, phylogenetically clustered plant groups may conform to abundant-centre patterns. Overall, we demonstrate that abundant-centre patterns are not a general ecological phenomenon; they tend to not apply to animals but can manifest in certain plant groups, depending on dispersal capabilities and evolutionary histories. Leveraging species' traits that account for dispersal improves models of abundant-centre patterns across geographic space.

Understanding patterns of biodiversity and the processes that drive them across varying spatial, temporal and taxonomic scales is a shared goal within macroecological and biogeographical research[1–3]. Such information is essential to improving our understanding of species' responses to environmental change, with applications to invasive species management[4,5] and conservation planning[6,7].

A common approach to studying macroecological processes and patterns is to develop and test ecogeographical "rules" or "hypotheses". As outlined by Baiser et al[8]., over the last decade, the study of ecogeographical rules has seen a resurgence due to increased data availability, methodological advancements and the need for applied research efforts, for example in invasive species management[5,9,10] and environmental change research[11,12]. Historically, many ecogeographical rules were developed to describe scaling relationships in species' diversity[13], body mass[14], range size[15], and geographical distributions of populations[16], such as the 'abundant-centre' hypothesis.

[1]School of Geography, University of Nottingham, Nottingham, UK. [2]Institute of Biology/Geobotany and Botanical Garden, Martin Luther University Halle-Wittenberg, Halle, Germany. [3]Royal Botanic Gardens, Kew, Richmond, Surrey, UK. [4]Section for Ecoinformatics & Biodiversity, Department of Biology, Aarhus University, Aarhus, Denmark. [5]German Centre for Integrative Biodiversity Research (iDiv) Halle-Jena-Leipzig, Leipzig, Germany. [6]Department of Biological, Geological and Environmental Sciences, University of Bologna, Bologna, Italy. ✉e-mail: connorpanter1301@gmail.com

The abundant-centre hypothesis derives from early ideas proposed by Grinnell[16], who likened the distribution and dispersal of animal populations to those of gas molecules occupying a vacuum[17,18]. The abundant-centre hypothesis posits that species' abundances are highest in their range centres and decline towards their range edges[17–24]. The hypothesis assumes that 1) a species' geographic range is a representation of its environmental or ecological niche[25] and 2) that environmental conditions are more optimal near the centre of a species' range and less optimal towards the range edges[18,26]. Since its inception, the abundant-centre hypothesis has been debated by macroecologists and biogeographers alike[23,27–30]. It has been tested within many taxonomic groups including birds[30–32], vascular plants[33–35], reef fishes[22,36,37], mammals[38–41], and coastal invertebrates[21,42–46]. It has also been tested across taxonomic groups[17,23,47]. Previous studies have employed an array of methodological approaches, such as focusing on centrality[23,48] as opposed to marginality[17,49].

Initially, the abundant-centre hypothesis has been tested across species' geographic space[19,21,22,44]. Support for the hypothesis remains scarce and recent research has shifted focus towards exploring abundance distributions across species' environmental or ecological niche space[29,30,39,41,48,50,51]. Specifically, Dallas et al.[23] tested the abundant-centre hypothesis across the geographical and the ecological niche spaces of 1400 species of North American birds, mammals, freshwater fishes and trees, concluding no consistent relationships between abundance and distance from range centre (hereafter 'abundance–distance relationships') in either case. Relatedly, Sporbert et al.[35] tested multiple macroecological rules, including the abundant-centre hypothesis, on 517 species of European vascular plants, suggesting that mixed support for abundant-centre patterns are likely driven by various environmental factors. It is often assumed that species' abundance is positively correlated with environmental suitability (an assumption of niche modelling), but this is far from always being the case[52]. To date, inconclusive empirical support for the abundant-centre hypothesis has been repeatedly found across geographic and ecological niche space[17,23,35,47]. It also remains unclear why abundant-centre patterns are rarely dominant (although others have concluded support[45,53]). Explanations may include differences in the methods used[17], interpretation of results or taxonomic groups studied, e.g., species that possess certain life histories such as juvenile planktonic life stages[46].

A major assumption of the hypothesis is that it assumes a species is in equilibrium with its environment, and that environmental conditions influencing abundance change gradually and monotonically from the range centre towards the range edge—as implied by Grinnell[16]. Similarly, it also assumes that conspecific populations are equally adapted to their environments, with recent research using species distribution models challenging this notion, instead finding that species demonstrate local adaptation across their ranges[54]. The hypothesis also assumes that optimal environments occur in the centre of a species' range and that the environment is strongly spatially autocorrelated. Accounting for these broad assumptions when testing the hypothesis, we would expect to find that species with higher dispersal capabilities, such as birds, larger-bodied mammals (e.g., large-bodied ungulates vs. small mammals) and plants with faster life cycles (e.g., non-woody vs. woody plants), may tend to follow abundant-centre patterns. These species may be better at tracking changing environments[55,56], maintaining higher abundances in their range centres where environmental conditions are assumed to be most favourable[57]. Given this, the underlying assumptions of equilibrial population dynamics across a species' geographic range assumes that population fluctuations are trivial, and that the species is not actively expanding its range.

Since the foundational work of Martínez-Meyer et al.,[48] exploration of abundant-centre patterns has shifted towards ecological niche space. However, no clear consensus has been reached within the scientific community as to which type of "space" is most appropriate to test this highly debated hypothesis, as the evidence supporting either perspective remains unconvincing. Calculating abundance–distance relationships in ecological niche space requires high-resolution data, to accurately capture the niche of the species at the micro-scale, which are often unavailable for most macroecological studies. Furthermore, the same environmental space would be required across all taxonomic groups, using the same environmental predictors, which may be more meaningful for some species than others. As such, fundamental research opportunities remain outstanding in geographic space, which can be explored with available large-scale species distribution data. One of many potential research avenues relates to linking species' ecological characteristics (species traits), as proxies for dispersal capabilities[58], to sampled abundance estimates.

Notably, multi-species analyses could be biased due to phylogenetic non-independence between taxa[59,60]. Lack of consideration and quantification of species' evolutionary relatedness limits our understanding of global abundance–distance relationships, resulting in calls for more robust studies incorporating appropriate comparative phylogenetic techniques[61]. To date, other than Dallas et al.[23], who explored the effects of body size and phylogenetic relatedness on support for the abundant-centre hypothesis, and Rivadeneira et al.[21], who linked evolutionary history and traits to porcelain crab (Porcellanidae spp.) abundance distributions, there have been no examinations of the effects of species' traits or phylogeny on support for the hypothesis across large taxonomic and spatial scales.

In this study, we use a trait-based and comparative phylogenetic approach to test the abundant-centre hypothesis across five major taxonomic groups: birds, mammals, freshwater fishes, reef fishes and plants. Given that previous research has suggested that support for the hypothesis may be scale-dependent[17,23], and abundance patterns might be closely related to species' dispersal capabilities[17,58,62], we also explore the effects of spatial scale and body size.

Based on previous research on animal distributions, we may expect to find differences between taxonomic groups, aligning with abundant-centre patterns, due to variation in dispersal capabilities between taxa[63]. Specifically, larger-bodied animals may be expected to conform more to such patterns as they are better able to track changing environments[64] (Table S1). Furthermore, omnivorous diets are known to affect dispersal capability, and thus we expect to find a dietary effect on abundance–distance relationships[64]. Dispersal capability in plants is also known to correlate with height, and to a lesser degree with seed mass[65], with taller species and those with smaller seeds being able to disperse further[66]. Therefore, we expect to find height and seed mass effects on plant abundance–distance relationships (Table S1). In line with r/K-selection theory, species towards the faster end of the fast-slow life-history continuum (i.e., r-selected species) may be better able to track changing environments due to their rapid population fluctuations, relative to K-selected species[67,68]. Similarly, based on previous research we expect to find effects of functional group[69], life span, and life form[70], on plant abundance–distance relationships (Table S1). Faster-growing species (e.g., grasses and herbs), and shorter-lived species (e.g., annuals and biennials), should be more likely to conform to abundant-centre patterns relative to their longer-lived and slower-growing congeners (e.g., trees). In addition, if abundant-edge distributions accurately represent invasion fronts—benefiting invasive species when dispersing[71]—then we may expect invasive species to not conform to abundant-centre patterns (Table S1). Dispersal capability is an important process that moderates a species' range size[72]; therefore, we expect that species with larger ranges are better adapters to the wider environment and will conform to abundant-centre patterns. Concurrently, terrestrial species distributed at higher latitudes tend to have larger ranges[73], and abundant-centre patterns may be more detectable at higher latitudes (Table S1). Finally, if abundant-centre patterns manifest themselves in species

**Table 1 | An overview of the sampling effort for this study**

| Study | Taxa | Compiled | | Used in analyses | |
|---|---|---|---|---|---|
| | | N species | N observations | N species | N observations |
| **Animals** | | | | | |
| Shalom et al.[22] | Reef fishes | 1235 | 3,066,573 | 1131 | 2,898,878 |
| Dallas et al.[23] | Birds, mammals and freshwater fishes | 1848 | 859,307 | 1716 | 793,217 |
| Santini et al.[17] | Birds and mammals | 104 | 4145 | 90 | 3694 |
| Freeman & Beehler[31] | Birds | 129 | 3225 | 113 | 2825 |
| Feldman et al.[121] | Birds | 6 | 312 | 4 | 208 |
| Martínez-Gutiérrez et al.[39] | Mammals | 1 | 72 | 1 | 72 |
| Wen et al.[40] | Mammals | 5 | 46 | 4 | 35 |
| Chaiyes et al.[41] | Mammals | 1 | 30 | 1 | 30 |
| Total (N animals) | | 3329 | 3,933,710 | 3060 | 3,698,959 |
| **Plants** | | | | | |
| Sporbert et al.[35] | Herbs, dwarf shrubs and shrubs | 532 | 2,059,847 | 480 | 1,654,725 |
| Dallas et al.[23] | Trees | 162 | 389,404 | 120 | 349,905 |
| Phiri et al.[81] | Dwarf shrub | 1 | 124 | | |
| McMinn et al.[122] | Herb | 1 | 42 | | |
| Dixon et al.[34] | Herb | 1 | 18 | | |
| Baer & Maron[123] | Herb | 1 | 11 | | |
| Gao et al.[124] | Tree | 1 | 5 | | |
| Total (N plants) | | 699 | 2,449,451 | 600 | 2,004,630 |
| Total (N total) | | 4028 | 6,383,161 | 3660 | 5,703,589 |

Table depicts the number of studies included, taxonomic group, number of species and observations used to calculate global abundance–distance relationships for 3660 species. Due to limited coverage of species trait data, some species were omitted prior to statistical analyses. N = sample size. Abundance–distance and trait data underlying this table are provided in Dryad at https://doi.org/10.5061/dryad.zgmsbccj2.

groups that possess similar dispersal capabilities through trait combinations, then we may expect to find abundant-centres nested within phylogenetically clustered taxa. To test this, we expect to find strong phylogenetic signals for abundant-centre conforming taxa but not for those that do not conform to the expected pattern.

Here, we use field observations of abundance to explore the effects of dispersal capability on species' abundance patterns using traits as proxies. We examine differences in conformity to abundant-centre patterns between taxonomic groups and test for the effects of spatial scale by compiling the following data for animals: 1) taxonomic group (categorical), 2) body size (cm; g), 3) invasive status (binary), 4) feeding guild (categorical), 6) extent (km), 7) grain ($km^2$) and 8) focus ($km^2$); and for plants: 1) functional group (categorical), 2) mean plant height (m), 3) invasive status (binary), 4) life span (categorical), 5) life form (categorical), 6) seed mass (mg), 7) extent (km), 8) grain ($km^2$) and 9) focus ($km^2$). To explore spatial patterns within global abundance–distance relationships, we also compiled geographic data for species-level range sizes ($km^2$) and calculated absolute latitudes (°) from range centroids.

We compute abundance–distance correlation coefficients (transformed to Fisher's Z values) which quantify the relationships between species' abundance and distance to range centroids from 14 published studies covering 3060 animal and 600 plant species. Collectively, we analyse 3,698,959 and 2,004,630 abundance observations for animals and plants, respectively (Table 1; Fig. 1a). The animal dataset includes 1683 bird species, 1,131 reef fish species, 202 mammal species and 44 freshwater fish species (Fig. 1a-d; Fig. 1f). The plant dataset comprises 386 herb species, 120 trees, 65 grasses and 29 shrubs (Fig. 1e). On average, there were 1209 ± 1944 ( ± SD) animal abundance observations (range: 5–11,948), and 3341 ± 5182 (range: 5–67,486) plant abundance observations (Table 1). Using separate weighted linear effects models to summarise the compiled correlation coefficients from abundance–distance relationships into grand mean and subgroup average Fisher's Z values, we test for the effects of

dispersal-related traits and range size variables on conformity to abundant-centre patterns (Table S2). To account for differences in Fisher's Z precision, we weighted each transformed correlation coefficient by the natural logarithm of the number of abundance observations.

Our results suggest that plants with better dispersal capabilities tend to conform to abundant-centre patterns. For animals, support for abundant-centre patterns is negligible across all taxonomic groups. Accounting for species' evolutionary histories improves models of abundance–distance relationships in geographic space but only for plants. Phylogenetically clustered taxa, such as non-woody plants, tend to conform to abundant-centre patterns. Evolutionary history did not explain abundance–distance relationships in animals, and thus evidence for abundant-centre patterns in animals remains weak.

## Results

### Abundance–distance relationships are relatively rare across taxonomic groups

Overall, global abundance–distance relationships differed between the animal and the plant data sets. Animals showed no clear abundant-centre patterns (Fig. 2a–d; Fig. 3a; grand mean of Fisher's $Z = 0.006 \pm 0.004$ (SE), $t = 1.447$, df = 3059, $p = 0.148$), with 47% ($n = 1430$; $n_{total} = 3060$) of animal species showing negative correlations between abundance and distance from the range centroid (indicative of an abundant-centre pattern) (Table 2). Of these negative correlations, 43% ($n = 620$ of 1430) were statistically significant, i.e., $p < 0.05$. Reef fishes and mammals generally did not conform to abundant-centre patterns, with 33% ($n = 370$ of 1131) and 50% ($n = 101$ of 202) of species within these groups having negative abundance–distance correlations, respectively (Fig. 2c, d; Table 2). Birds and freshwater fishes showed some support for abundant-centre patterns. However, these between-group differences were marginal, with 56% ($n = 936$ of 1683) and 52% ($n = 23$ of 44) of species showing negative abundance–distance correlations, respectively (Fig. 2a, b; Table 2).

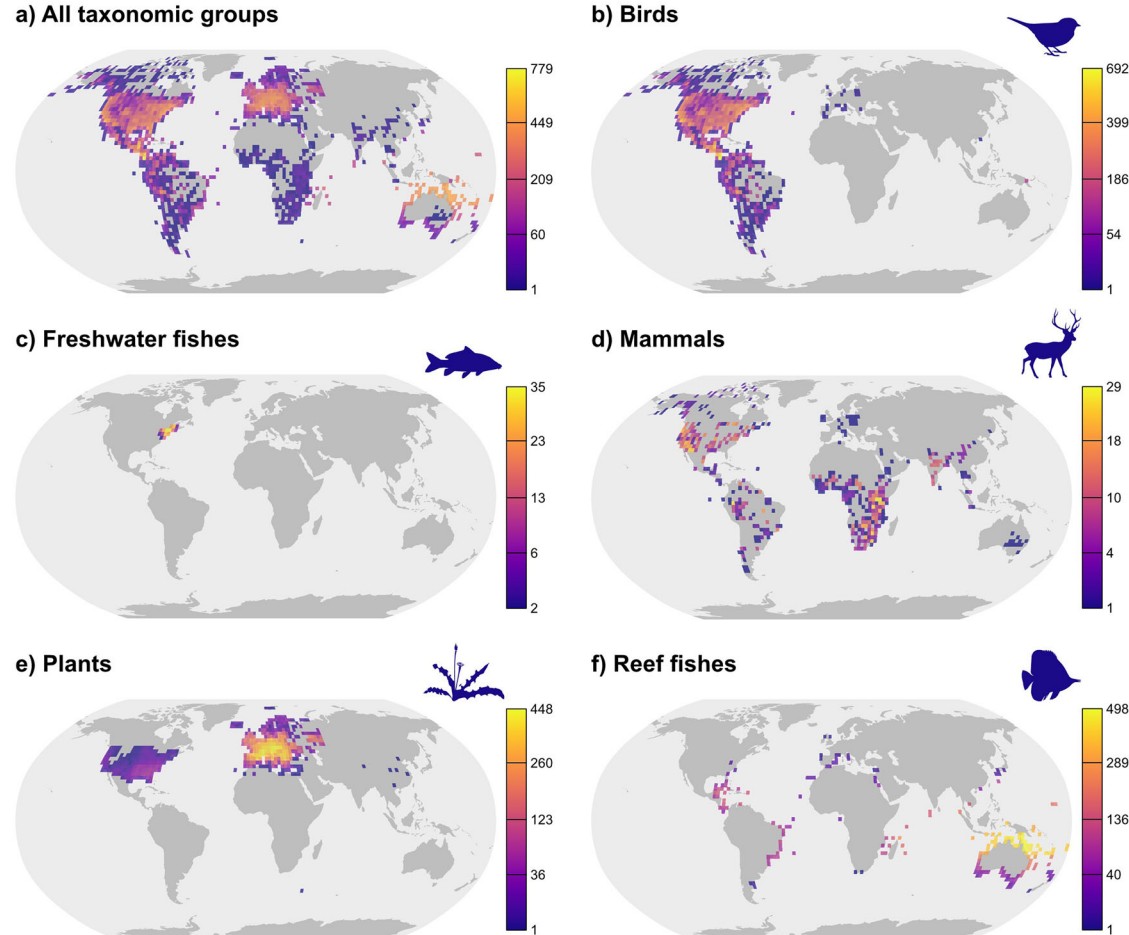

**Fig. 1 | Spatial distribution of 5,703,589 abundance observations for 3660 species compiled this study. a** all taxonomic groups combined, **b** birds, **c** freshwater fishes, **d** mammals, **e** plants and **f** reef fishes. Data for 3060 animal and 600 plant species were subsequently used for statistical analyses. Maps represent the number of species with abundance data per 2.5° grid cell (for display purposes only), reprojected into the Robinson projection. Values range from dark purple (low number of species) to light yellow (high number species), with breaks obtained using a square-root transformation. Data underlying this figure are provided in Dryad at https://doi.org/10.5061/dryad.zgmsbccj2.

Plants generally showed some consistency with the abundant-centre hypothesis (Fig. 2e–h; Fig. 3b; grand mean of Fisher's $Z = -0.08 \pm 0.005$ (SE), $t = -15.7$, $p < 0.001$), with 75% ($n = 446$ of 600) of plant species having negative correlations between abundance and distance from the range centroid (Table 2). Of these negative correlations, 75% ($n = 332$ of 446) were statistically significant. Grasses, herbs and shrubs tended to follow abundant-centre patterns with 82% ($n = 53$ of 65), 84% ($n = 332$ of 386) and 62% ($n = 18$ of 29) of species possessing negative abundance-distance correlations, respectively (Fig. 2e-g; Table 2). Trees were the only plant subgroup inconsistent with abundant-centre patterns, with only 44% ($n = 53$ of 120) of species showing negative abundance-distance correlations (Fig. 2h; Table 2).

**Accounting for species traits and range size can improve models of abundant-centre patterns across geographic space**

We used a stepwise model selection approach to test the significance of different predictor variables for the distribution of Fishers Z scores for animal (i.e., taxonomic group, invasive status, body mass, feeding guild, absolute latitude and $log_{10}$ range size) and plant species (i.e., functional group, invasive status, life form, absolute latitude, and $log_{10}$ seed mass). For animals, the most parsimonious model included the taxonomic group and $log_{10}$ transformed range size with predicted negative abundance–distance relationships for birds, positive relationships for mammals and reef fishes, and a positive effect of $log_{10}$ range size on abundance–distance relationships (Fig. 3a; Table S3).

For plants, the most parsimonious model included the functional group, invasive status, absolute latitude and $log_{10}$ transformed seed mass. This model predicted negative abundance–distance relationships for grasses and herbs, more negative relationships for invasive versus non-invasive plants and a negative effect of $log_{10}$ seed mass and absolute latitude on abundance–distance relationships (Fig. 3b; Table S3).

Repeating this stepwise variable selection approach for the different taxonomic groups yielded different sets of significant predictors. For birds and freshwater fishes, feeding guild was the only significant predictor indicating that herbivorous species showed more negative abundance-distance relationships than carnivores or omnivores (Table S4). For reef fishes, there was a significant negative effect of absolute latitude on abundance–distance relationships (Table S4). However, there were no significant predictors of abundance–distance relationships for mammals (Table S4). Within plant taxa, grasses and herbs showed more negative relationships at higher latitudes, whereas a higher number of negative relationships were found for invasive shrubs relative to non-invasive species, and for those with larger seed mass (Table S4). For trees, none of the predictors were significantly related to the observed differences in abundance–distance relationships (Table S4).

**Interactions between species traits and geographic variables**

Separately within the animal and the plant data sets, we explored whether linear models could be improved by incorporating bivariate

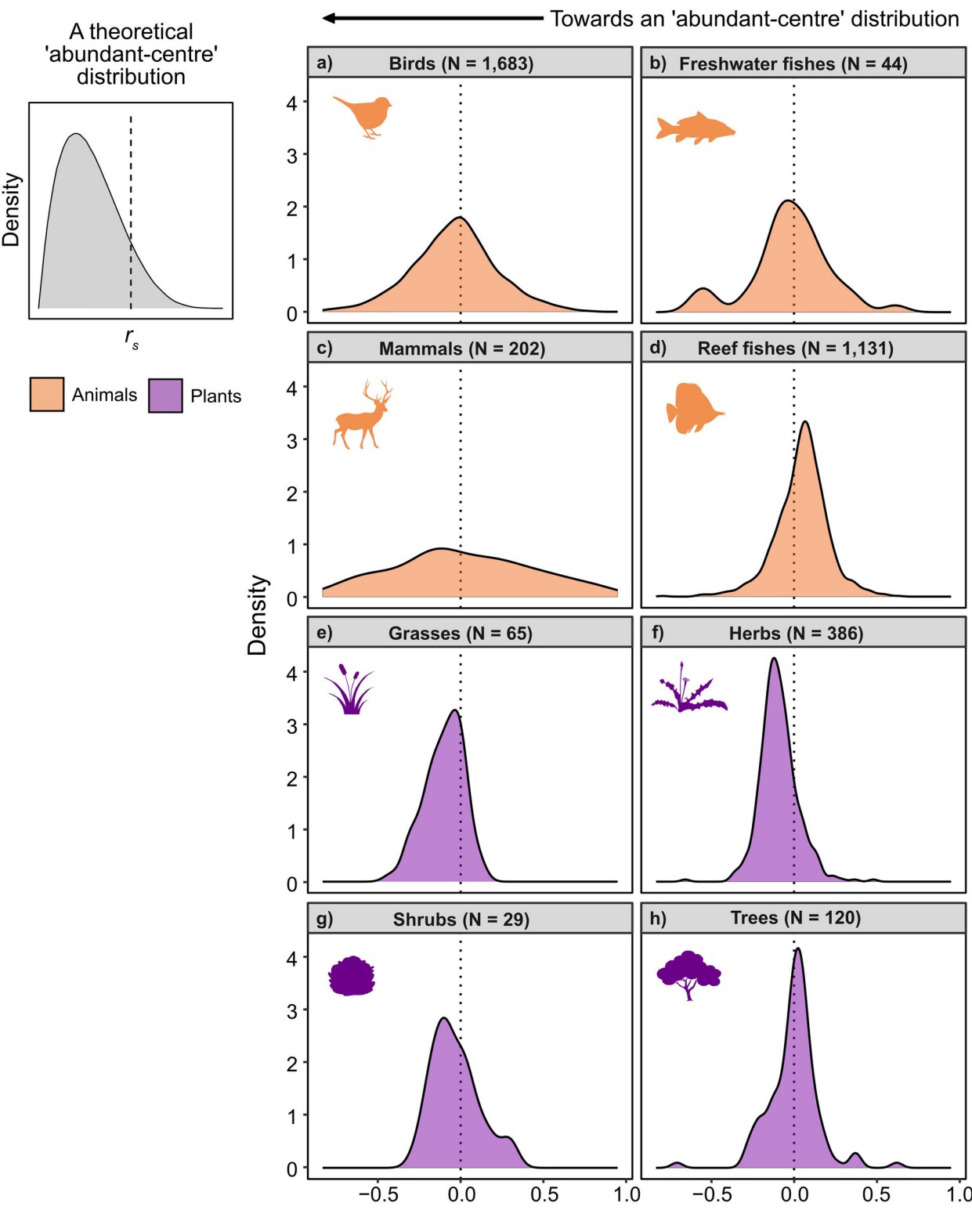

interactions between predictor variables. For each data set, we calculated all fixed effect combinations and bivariate interaction terms and selected the most important interactions based on the Akaike Information Criterion (AIC) (Burnham & Anderson 2002). For animals, there were important interaction effects between taxonomic group and feeding guild (group × feeding guild: $F_{1,6} = 3.875$, $p < 0.001$), body size (group × body size: $F_{1,3} = 4.251$, $p = 0.005$) and latitude (group ×

latitude: $F_{1,3} = 12.111$, $p < 0.0001$) (Fig. 4a–c; Table S5). These interactions indicated that omnivorous mammals and reef fishes more closely approached the expected abundance–distance pattern (while being non-significant compared to herbivorous or carnivorous species which had significantly positive relationships) (Fig. 4a; Table S5). Larger-bodied mammals were more likely to conform to abundant-centre patterns (Fig. 4b), as were mammals and freshwater fishes from higher

**Fig. 2 | Kernel density plots for global abundance–distance correlations for 3660 animal and plant species.** Abundance–distance relationships were calculated using Spearman's Rank Correlation Coefficients ($r_s$) visualised for animal taxonomic group (birds, freshwater fishes, mammals and reef fishes) and plant functional group (grasses, herbs, shrubs and trees). An 'abundant-centre' distribution would be expected to feature predominantly negative $r_s$ values (illustrated in top-left panel). **a** Kernel density distribution of abundance–distance relationships for 1683 bird species, **b** Kernel density distribution of abundance–distance relationships for 44 freshwater fish species, **c** Kernel density distribution of abundance–distance relationships for 202 mammal species, **d** Kernel density distribution of abundance–distance relationships for 1131 reef fish species, **e** Kernel density distribution of abundance–distance relationships for 65 grass species, **f** Kernel density distribution of abundance–distance relationships for 386 herb species, **g** Kernel density distribution of abundance–distance relationships for 29 shrub species, and **h**) Kernel density distribution of abundance–distance relationships for 120 tree species. Source data are provided as a Source Data file.

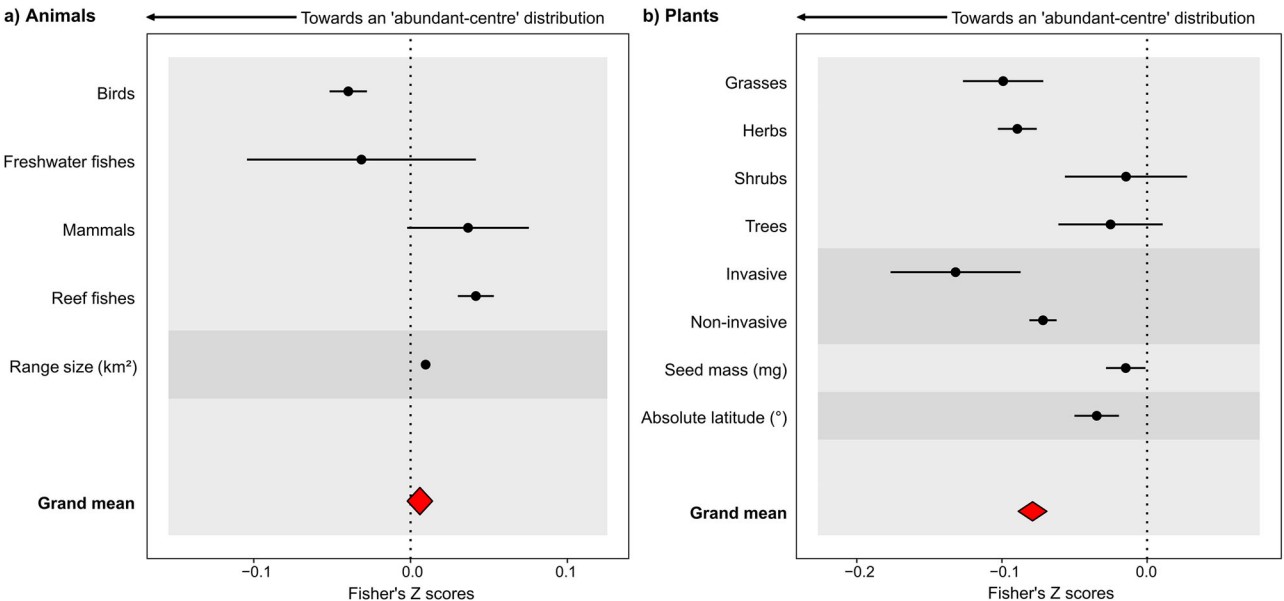

**Fig. 3 | Modelled effects of significant species traits and geographical variables on abundance–distance relationships (Fisher's Z) for 3060 animal species and 600 plant species, by predictor group. a** Effects of species traits and geographical variables for 3060 animal species, and **b** effects of species traits and geographical variables for 600 plant species. Negative Fisher's Z scores (left of dotted line) indicate an 'abundant-centre' distribution effect. Group-level Z values were derived from marginal mean estimates for categorical predictors (animals: Taxonomic group, Invasiveness and Feeding guild; plants: Functional group, Invasive and Life Form) and from regression relationships for continuous predictors (animals: $\log_{10}$ Body size (cm; g), Absolute latitude (°) and $\log_{10}$ Range size (km²); plants: $\log_{10}$ Mean plant height (m), Absolute latitude (°), $\log_{10}$ Range size (km²) and $\log_{10}$ Seed mass (mg), each scaled to unit variance) (see Table S4 for interaction model outputs). Error bars represent approximated 95% confidence intervals. Source data are provided as a Source Data file.

latitudes (Fig. 4c; Table S5). For plants, there was a significant interaction effect between functional group and seed mass (functional group × seed mass: $F_{1,3} = 6.483$, $p < 0.001$) (Fig. 5a; Table S5), with shrubs that have larger seeds tending to support abundant-centre patterns, whereas invasive species with larger seed mass tended not to conform with this pattern (Fig. 5b; Table S5).

### Phylogenetic signals in abundance–distance relationships

Accounting for phylogenetic relatedness did not improve the model fit for the animal data set (Table 3). Conversely, for plants, incorporating the phylogenetic correction matrix with the observed value of Pagel's λ (-0.2) significantly improved the model fit and removed the significance of the estimated grand mean effect (Fisher's Z = −0.04, $t = −0.98$, $p = 0.33$). Similarly, our model residuals (tested separately for the bird, fish, mammal, and plant data sets) were mostly independent of species phylogenetic relatedness – again, except for the calculation of the grand mean effect in the plant data set (Table 3).

## Discussion

Across what is, to our knowledge, the most extensive global dataset on animal and plant species abundances to have been used for testing the abundant centre hypothesis, we observed that global abundance–distance relationships followed taxon-specific patterns, often including both positive and negative values. Our findings bring some clarity to previous research which yielded little to no support for

the abundant-centre hypothesis[17,23,34,36,46,53,74]. However, in contrast to these studies, we observed that some plant groups appeared to follow abundant-centre patterns, whereas animals did not show any strong tendency toward such patterns. Inclusion of dispersal-related species traits and evolutionary histories improved some of our models of abundance–distance relationships. We expected to find conformity to abundant-centre patterns in species with higher dispersal capabilities. We found that dispersal capability does not explain abundance–distance relationships in animals but likely explains such patterns in non-woody plants, which may be better at tracking changing environments due to shorter generation times and higher dispersal abilities. Our findings suggest that plant species less limited by dispersal constraints are more likely to follow abundant-centre patterns in geographic space. Inclusion of phylogenetic relatedness in abundance–distance relationships returned a phylogenetic signal for certain taxonomic groups, indicating that abundance-distance relationships are not a general phenomenon but are restricted to phylogenetically clustered taxa.

### Underlying assumptions may explain why global abundance–distance relationships remain relatively rare

Fundamental assumptions of the abundant-centre hypothesis are that a species is in equilibrium with its environment and that environmental conditions influencing abundance change gradually and monotonically from the range centre towards the range edge. Violation of

**Table 2 | Spearman Rank Correlation Coefficients ($r_s$) exploring relationships between abundance and distance from geographic range centroids for 3060 animal and 600 plant species**

| Predictors/subgroups | N spp. (%/taxonomic group) | % total data set | $r_s$ < 0 (% subgroup)[a] | % $p$ < 0.05 | $r_s$ ≥ 0 (% subgroup)[a] | % $p$ < 0.05 |
|---|---|---|---|---|---|---|
| Animals | 3060 (100) | 83.3 | 1430 (46.7) | 43.4 | **1630 (53.3)** | 49.4 |
| Group | | | | | | |
| Birds | 1683 (55) | 45.8 | **936 (55.6)** | 35.3 | 747 (44.4) | 25.4 |
| Freshwater fishes | 44 (1.4) | 1.2 | **23 (52.3)** | 34.8 | 21 (47.7) | 23.8 |
| Mammals | 202 (6.6) | 5.5 | 101 (50) | 34.7 | 101 (50) | 35.6 |
| Reef fishes | 1131 (37) | 30.8 | 370 (32.7) | 21.8 | **761 (67.3)** | 75.6 |
| Invasive status | | | | | | |
| Invasive | 41 (1.3) | 1.1 | **21 (51.2)** | 28.6 | 20 (48.8) | 60 |
| Non-invasive | 3019 (98.7) | 82.1 | 1409 (46.7) | 43.6 | **1610 (53.3)** | 49.3 |
| Feeding guild | | | | | | |
| Carnivores | 1436 (46.9) | 39.1 | 619 (43.1) | 44.9 | **817 (56.9)** | 53.1 |
| Herbivores | 610 (19.9) | 16.6 | 288 (47.2) | 42 | **322 (52.8)** | 49.1 |
| Omnivores | 1014 (33.1) | 27.6 | **523 (51.6)** | 42.3 | 491 (48.4) | 43.6 |
| *Plants* | 600 (100) | 16.4 | **446 (74.3)** | 74.6 | 154 (25) | 40.9 |
| Functional group | | | | | | |
| Grasses | 65 (10.8) | 1.8 | **53 (81.5)** | 69.8 | 12 (18.5) | 25 |
| Herbs | 386 (64.3) | 10.5 | **322 (86)** | 81.6 | 64 (16.6) | 40.6 |
| Shrubs | 29 (4.8) | 0.8 | **18 (62.1)** | 77.8 | 11 (37.9) | 63.6 |
| Trees | 120 (20) | 3.3 | 53 (44.2) | 34 | **67 (55.8)** | 40.3 |
| Invasive status | | | | | | |
| Invasive | 23 (3.8) | 0.6 | **20 (87)** | 95 | 3 (13) | 0 |
| Non-invasive | 577 (96.2) | 15.8 | **426 (73.8)** | 73.7 | 151 (26.2) | 41.7 |
| Life form | | | | | | |
| Geophytes | 20 (3.3) | 0.5 | **16 (80)** | 81.3 | 4 (20) | 25 |
| Hemicryptophytes | 256 (42.7) | 7 | **218 (85.2)** | 80.3 | 38 (14.8) | 26.3 |
| Phanerophytes | 186 (31) | 5.1 | **99 (53.2)** | 52.5 | 87 (46.8) | 43.7 |
| Therophytes | 138 (23) | 4.2 | **113 (83.7)** | 81.3 | 25 (16.3) | 56 |
| Life span | | | | | | |
| Annuals | 66 (11) | 1.8 | **51 (77.3)** | 78.4 | 15 (22.7) | 46.7 |
| Annuals/Biennials | 10 (1.7) | 0.3 | **9 (90)** | 88.9 | 1 (10) | 0 |
| Biennials | 21 (3.5) | 0.6 | **18 (85.7)** | 72.2 | 3 (14.3) | 0 |
| Biennials/Perennials | 9 (1.5) | 0.2 | **5 (55.6)** | 80 | 4 (44.4) | 25 |
| Perennials | 494 (82.3) | 13.5 | **363 (73.5)** | 73.8 | 131 (26.5) | 42 |

[a]Bold values show abundance–distance relationships that exceed more than 50% of the predictor group.

Negative $r_s$ values, i.e., $r_s$ < 0, indicate 'abundant-centre' patterns where species abundance is highest in the geographic range centre and declines towards the range edges. $r_s$ ≥ 0 indicates an inverse 'abundant-edge' pattern. Correlation coefficient weights in relation to the entire data set are also presented. Percentage of statistically significant correlations ($p$ < 0.05) also presented. Correlation coefficient summaries only shown for categorical predictors. Data underlying this table are provided in Dryad at https://doi.org/10.5061/dryad.zgmsbccj2.

these assumptions may explain why global abundance–distance relationships were relatively rare across most taxonomic groups. It may be even more challenging to detect abundant-centre patterns in species with fine-scale habitat selection, where changes in structural heterogeneity, orography, aspect, soil type, microclimate and availability of cover structures influence a species' abundance throughout time and space.

A recent study found general support for abundant-centre patterns in birds[57], despite other studies suggesting that these patterns are rarely detected in this group[17,23]. We failed to detect a strong pattern in our bird data. Wing morphology has been shown to be an important predictor of dispersal distance in birds[75,76]. To account for this, we conducted a supplementary analysis exploring the effects of the Hand-Wing Index (HWI) on global abundance–distance relationships for 1,683 bird species (see Supplementary Methods). We extracted HWI values from the AVONET database[77] and found no significant HWI effect on modelled bird abundance–distance relationships ($F_{1,1681}$ = 0.0004, $p$ = 0.986, $R^2$ = < 0.001; Table S9). This suggests

that dispersal capability in birds does not explain abundance–distance relationships across geographic space.

As predicted, invasive animals did not show support for abundant-centre patterns; however, they also did not show significantly positive abundance–distance relationships, which might be expected of invasion fronts[78]. Unlike other studies[23,35], our test of the abundant-centre hypothesis was global in its approach. Prior to this study, the largest existing test of the hypothesis found weak support for abundant-centre patterns in geographic and ecological niche space[23]. The authors also attempted to explain variation in abundance–distance relationships by exploring the effects of body size, range size and climatic niche area[23]. We decided not to use the range size variable calculated by Dallas et al.,[23] who interpreted minimum convex polygons (MCPs) around sampling points as proxies for a species' range, and instead sourced most of our range size estimates from published IUCN expert range maps[79]. Only where expert range maps were unavailable, we used MCPs for some terrestrial species with small ranges, i.e., understudied species. It must be noted that MCPs have a coarse outer edge at low sampling intensities

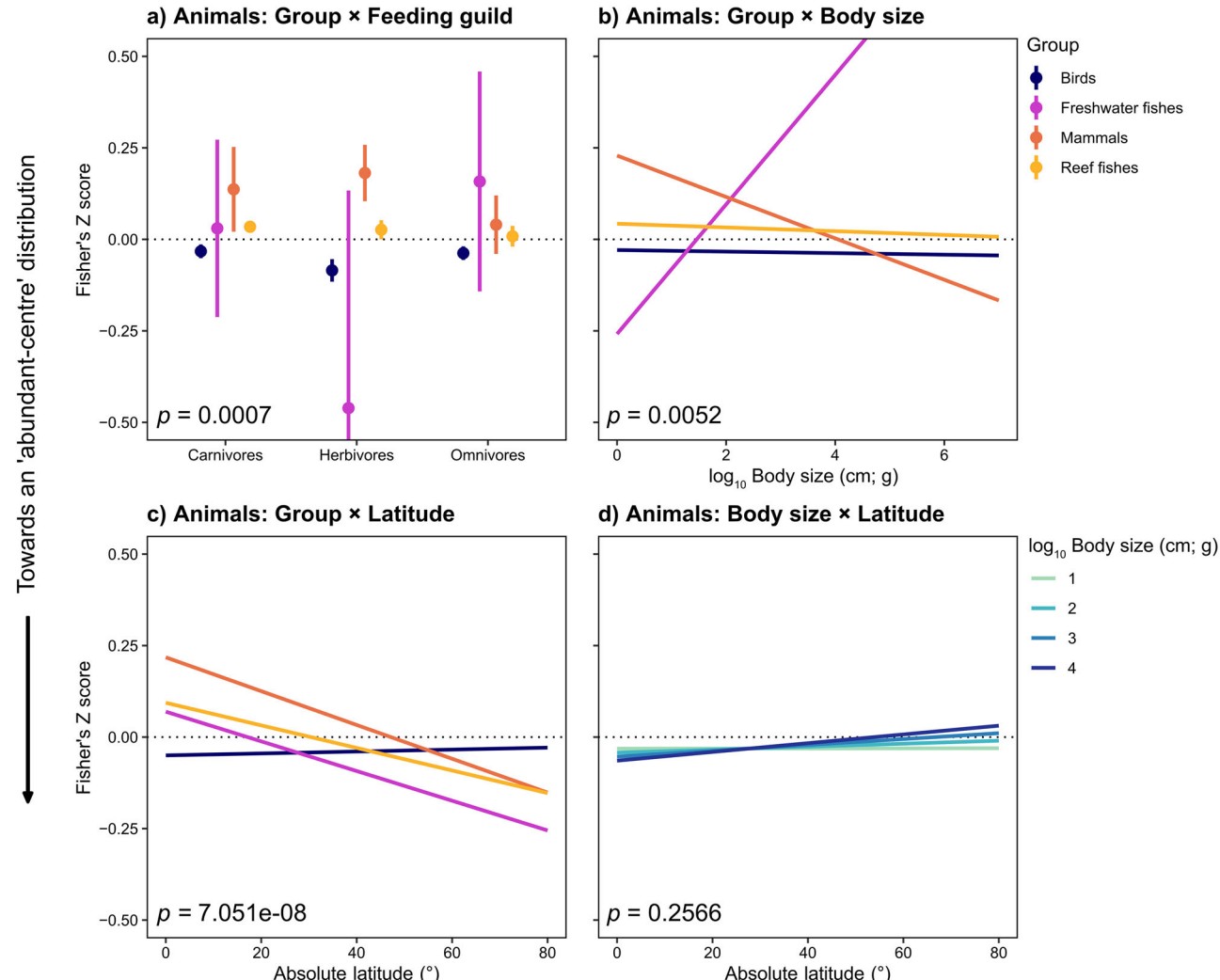

**Fig. 4 | Important interaction effects of species traits on abundance–distance relationships (Fisher's Z) for 3060 animal species.** Negative Fisher's Z scores (left of dotted line) indicate an 'abundant-centre' distribution effect, derived from a linear model weighted by the number of abundance observations per species. Interactions modelled as marginal joint effects (Table S5) for **a** Group × Feeding guild, **b** Group × $\log_{10}$ Body size (cm; g), **c** Group × Absolute latitude (°) and **d** $\log_{10}$ Body size (cm; g) × Absolute latitude (°). **a** Error bars represent 95% confidence intervals. Source data are provided as a Source Data file.

and are therefore sensitive to outliers[80], and sampling efforts only cover most of a species' range in exceptional circumstances (such as thorough field surveys of endemic species on small islands[81]). Furthermore, Dallas et al.[23] used eBird data to test range-wide abundant-centre patterns in North American birds, which may introduce challenges associated with sampling procedures when calculating abundance–distance relationships[57]. To ensure that our approach was as comprehensive as possible, we also included the largest data set on birds from Dallas et al.[23]; however, we are aware that caution must be used when interpreting the results because of the known issues with the use of presence-only data. Traits included in Dallas et al.[23]—body size, range size and climatic niche area − explained very little variation within their models (birds: $R^2 = 3\%$; trees = $R^2 = 3\%$, mammals: $R^2 = 4\%$ and freshwater fishes: $R^2 = 2\%$), whereas our most parsimonious trait interactions models explained 5% of model heterogeneity for animals and 16% for plants. An explanation for this may simply lie in the number of dispersal-related traits used to test the hypothesis or the choice of methods used (more robust analytical techniques, which may be more appropriate when testing broad-scale relationships between species abundances and their environments[17,50,52].

The use of species traits to explore global abundance–distance relationships enabled strong signals to be detected within a

comparatively large data set. Compared to animals, our trait-based method appeared to work better when explaining support for abundant-centre patterns in plants, suggesting that other processes that remain untested here may contribute to the unexplained heterogeneity in observed abundance–distance patterns. Biotic and abiotic processes may include, but are not limited to, interspecific interactions[82], spatiotemporal patterns in resource availability[83], climate and environmental suitability[84], geodiversity[85], and pressures from human activities[86]. Failure to account for phylogeny in ecological regression models may violate independence assumptions[87], although previous evidence suggests that phylogeny has no broad effect on modelled abundance–distance relationships at the species level[23]. Here, we demonstrate that incorporating species' evolutionary histories in models of abundance–distance relationships can improve model robustness and generalizability. However, this effect appears to depend strongly on the taxonomic group studied, and was only significant in plant species for which common traits through shared ancestries might jointly influence the strength of the abundance–distance relationships. Pagel's λ assumes that traits follow a Brownian motion model of evolution[88] which is likely a broad oversimplification (but may also explain some of the observed patterns in our dataset). Furthermore, it should be note that the metric used to

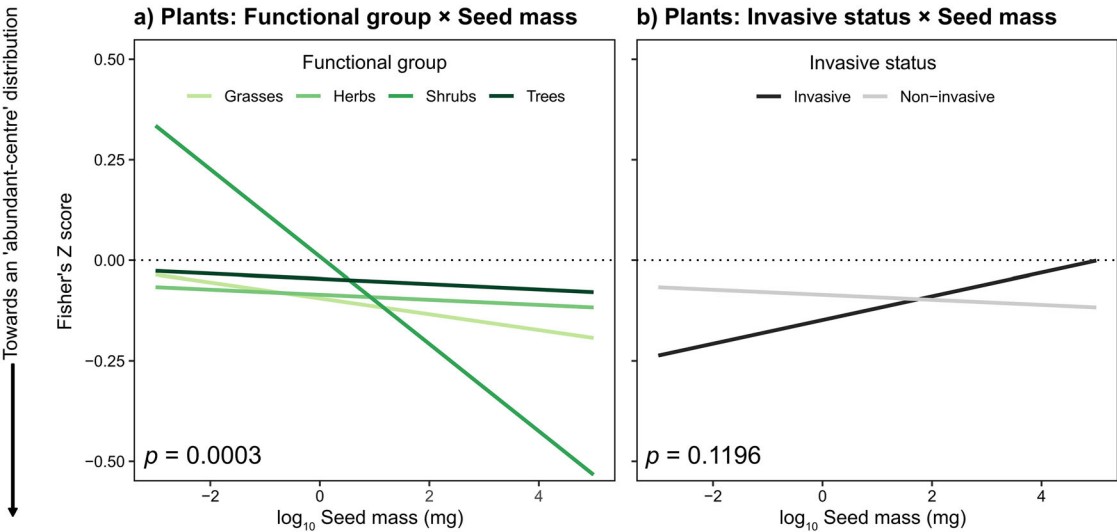

**Fig. 5 | Important interaction effects of species traits on abundance–distance relationships (Fisher's Z) for 600 plant species.** Negative effects (left of dotted line) indicate an 'abundant-centre' distribution effect, derived from a linear model weighted by the number of abundance observations per species. Interactions modelled as marginal joint effects (Table S5) for **a** Functional group × $\log_{10}$ Seed mass (mg) and **b** Invasive status × $\log_{10}$ Seed mass (mg). Source data are provided as a Source Data file.

## Table 3 | Tests for phylogenetic signals in abundance–distance relationships for birds, freshwater and reef fishes, mammals, and plants

**a) Goodness of fit measures for intercept-only models with different values for Pagel's λ.**

| Species group | $AIC_{\lambda=0}$[a] | $AIC_{\lambda=1}$[a] | $AIC_{\lambda fit}$[a] | $\lambda_{fit}$ estimate[a] |
|---|---|---|---|---|
| Birds | 484.3 | 2251 | 484.3 | <0.0001 |
| Fishes | −813.4 | 241.5 | −813.4 | <0.0001 |
| Mammals | 318.5 | 510.8 | 318.5 | <0.0001 |
| Plants | −664.9 | 64.2 | **−683.3** | **0.177***** |

**b) Test for phylogenetic signals in residuals of models with increasing complexity.**

| Species group | $\lambda_{grand\ mean}$[a] | $\lambda_{fixed}$[a] | $\lambda_{interactions}$[a] |
|---|---|---|---|
| Birds | <0.0001 | <0.0001 | <0.0001 |
| Fishes | <0.0001 | <0.0001 | <0.0001 |
| Mammals | <0.0001 | <0.0001 | <0.0001 |
| Plants | **0.177***** | <0.0001 | <0.0001 |

[a]Bold values indicate statistically significant phylogenetic signal effects.
a) shows the Akaike Information Criterion (AIC) values for intercept-only generalised least-squares models (estimated with maximum likelihood) including a phylogenetic correlation structure with Pagel's λ either set to zero (λ0; no phylogenetic signal), set to one (λ1; full phylogenetic signal) or set to the observed phylogenetic signal for Fisher's Z values (λfit). b) shows phylogenetic signal statistics (based on Pagel's λ) and respective *p*-values for the residuals from weighted linear effects models that either included only the intercept ($\lambda_{grand\ mean}$), the set of significant fixed effects from the most parsimonious models ($\lambda_{fixed}$, c.f., Fig. 3), or the important interactions and their fixed effects ($\lambda_{interactions}$, c.f., Figs. 4 and 5). Data underlying this table are provided in Dryad at https://doi.org/10.5061/dryad.zgmsbccj2.
*** *p* < 0.001.

calculate phylogenetic signal in our data set will likely influence the results obtained[89].

Our focus on species' dispersal capabilities responds to recent research[58] which found dispersal to be an important process driving abundance–distance patterns within simulated environments. Other research has found that species that are better dispersers tend not to conform to abundant-centre patterns[46]. However, Ntuli et al.[46] tested the abundant-centre hypothesis on intertidal species with planktonic life stages, and direct comparisons with our findings may not yield

ecologically meaningful conclusions. In the context of the taxonomic groups included in our analysis, our use of dispersal-related species traits suggests that dispersal capabilities should be accounted for when testing macroecological hypotheses such as the abundant-centre hypothesis.

Similar to other tests of the abundant-centre hypothesis, our sampling approach involves caveats. One limitation relates to the underlying abundance database used, which inherently vary in different sampling protocols, time periods studied, study sites and taxonomic focus. We explored the effects of underlying databases within each study on abundance–distance relationships for animals and plants as part of our supplementary analyses (see Supplementary Methods). There were significant effects of the underlying database in both animal ($F_{1,3051} = 14.364$, $p < 0.0001$) and plant ($F_{1,598} = 66.302$, $p < 0.0001$) data sets, contributing 3% and 10% of variation within our animal and plant abundance–distance relationships, respectively (Table S6; Table S7). These signals may simply be an artifact of the species included within each data set, as duplicated species data between studies were removed prior to analysis and abundance data from the study with the higher number of observations was retained. In addition to this, most species' geographical ranges are likely to have been under-sampled, and abundance estimates are mostly not derived from the full extents of their ranges.

Our test of the hypothesis assumes that conspecific populations are equally adapted to the environment. However, there is local adaptation across some species' ranges[54], which we were unable to account for using our approach. Species that have undergone historical range contractions, either naturally or from human activities, may become ecologically marginalised[90], which may introduce error into our assessments of abundance–distance relationships. We attempted to account for this by exploring how abundance–distance relationships manifest in species that have undergone historical range contractions. Using historical range data from the PHYLACINE database[91] (version 1.2), we found no significant effect of historical range contractions on global abundance–distance relationships in 202 mammal species ($F_{1,200} = 1.242$, $p = 0.266$, $R^2 = 0.001$; Table S10). Despite not finding a significant effect of historical range contractions on abundance–distance relationships, it is important to consider that our study investigates natural processes using data from a human-modified world. Future research should leverage available historical

range data across multiple taxonomic groups and explore, in detail, the effects of natural and human historical range dynamics on abundance–distance relationships.

Sampling efforts are likely to be influenced by differences in data quantity and quality between taxa. For example, bird abundance data are more readily available from open-source repositories (such as eBird – ebird.org) than abundance data for most other taxa. We attempted to account for this by weighting the linear models by the natural logarithm of species sample size, such that better-sampled species were assigned larger weights in our analyses than less-sampled species. Even though we employed a global approach to testing the abundant-centre hypothesis, our study is still bound by the geographical constraints of the studies included within our data set. This is demonstrated by the geographical biases within our data set (see Fig. 1), which may influence our findings. There were notable taxonomic gaps within our data set, and a skew towards more commonly sampled groups. During our literature searches, we were unable to obtain any abundance estimates for reptiles, amphibians, marine mammals, pelagic fishes, corals, fungi and most invertebrate groups.

Despite these limitations, we were able to calculate abundance–distance relationships for 4036 species, with 3,660 used in our analyses. Due to the volume of species data included in this study, we were unable to control for spatiotemporal differences between migratory and non-migratory species abundances, which have been shown to influence support for the abundant-centre hypothesis[30]. However, previous research has suggested otherwise[23,47]. Our supplementary analyses on a data set of 1,683 bird species suggest that migratory strategy has no effect on global abundance–distance relationships ($F_{1,1680} = 0.526$, $p = 0.591$, $R^2 = < 0.001$; Table S8); this aspect warrants further attention by future studies. We focused on testing the hypothesis by a measure of centrality as opposed to marginality. Depending on the shape of a species' geographic range, such measures can differ largely as occurrences far from the range centroid can also be far from the range edge[17]. All of the studies included in our analyses specifically tested the abundant-centre hypothesis in some form. Relatedly, preference towards the publication of studies that find positive support for abundant-centre patterns over those that do not may be apparent, which should be taken into consideration when interpreting our findings. Finally, we searched for studies published in English, French and Spanish, but did not obtain any relevant search results in languages other than English, which is a limitation of our results. We suspect this is due to the search terms being optimised for literature published in English.

Our study indicates that accounting for dispersal-related traits and phylogeny improves models of abundant-centre patterns across geographic space. Future tests of the abundant-centre hypothesis should aim to (i) include data for under-represented taxonomic groups, such as reptiles, amphibians and fungi, to enhance our overall understanding of broad-scale patterns of species' abundances, and (ii) compare relationships between abundant-centre patterns and habitats at the micro-scales. Finally, we recommend that future tests of the abundant-centre hypothesis attempt to examine other processes, such as interspecific interactions and impacts of human activities, which may underlie abundance–distance relationships and account for additional unexplained heterogeneity in our data set.

## Methods

### Literature searches

We conducted a systematic literature search on 23rd July 2021 by querying the ISI Web of Science database (apps.webofknowledge.com) with the following search string for an initial broad search: "(abundan* OR abundance-cent* OR abundant niche-cent* OR niche cent* OR abundant-centre hypothesis) AND (range OR geographic range OR range size OR range edge OR species distribution)" using the TITLE field. We retained all studies that 1) comprised peer-reviewed primary studies, 2) presented globally extensive abundance point observations across all taxonomic groups, 3) were published between 1990 and 2020, 4) included extractable data relating to observed/estimated abundance counts and 5) were published in English, French or Spanish language.

Examination of the returned studies ($N = 818$) revealed that some key literature was missing from our results. Therefore, we used the studies returned from our initial search as reference sources for additional key search terms to derive an optimised search string using the *litsearchr* R package[92]. Search terms were extracted from unique study titles, abstracts and tagged keywords (e.g., terms such as 'range edge', 'abundance' and 'species range'). We built a keyword co-occurrence network and quantitatively assessed potential search terms using a 60% cumulative cut-off point[92]. Resulting search terms ($N = 326$) were grouped into either 1) the 'species group', 2) 'geographic group' or 3) 'both groups' depending on whether the term referred to a species concept or geographic concept. Grouping refers to the string of search terms either side of the Boolean operator 'AND'. We manually removed irrelevant search terms ($N = 268$; e.g., 'field sites', 'habitat patch' and 'statistically significant') and retained the most relevant search terms ($N = 58$) which formed our optimised search string (Table S11). To verify whether our resulting search string was fully optimised, we cross-referenced four key articles that we expected to be included in the optimised search results[23,34,38,44]—all of which were included. We queried the Web of Science database using our optimised search string and obtained 531 studies. After screening of titles and abstracts, we retained 23 studies for data extraction.

We supplemented our Web of Science literature search with the studies included within the foundational synthesis by Sagarin & Gaines[19]. We then conducted a snowball search of the literature that cited Sagarin & Gaines[19] up until 31st December 2020. This resulted in a combined literature database of 1109 studies. Due to non-conformity to our selection criteria, we excluded 1000 studies after screening titles and abstracts and an additional 95 studies after screening of full texts, leaving us with 14 studies that were suitable for data extraction.

### Data extraction and processing

From each study, and for each species, we extracted raw abundance values and distance from the species' geographic range centroids (in km). Corresponding authors were contacted via email correspondence if data were not publicly available. Where data were not publicly available and the corresponding author was unable to provide the data, we extracted abundance and distance data from appropriate figures within the published articles using the web-based tool Web-PlotDigitizer (version 4.5)[93]. If abundance data were available but distance values were not, global range maps were obtained in shapefile formats for each species and range centroids were calculated. We obtained global range shapefiles for terrestrial mammals from the IUCN Red List[79] and for birds from the BirdLife Data Zone (version 2020.1)[94], thus accounting for the entire ranges for migratory and non-migratory species. IUCN range maps have been criticized due to oversimplification of species' ranges derived from sampling bias[95] but represent the most comprehensive spatial data set available for our study species. If global polygon range maps for particularly under-sampled taxonomic groups were unavailable, e.g., invertebrates and some plant species (N spp. = 8), we downloaded species occurrence point data from the Global Biodiversity Information Facility (https://www.gbif.org). Occurrence data were then cleaned using the *CoordinateCleaner* package[96] and manual checks were performed to remove any remaining outliers[97]. We calculated minimum convex hulls for terrestrial species, in an attempt to not overestimate their global range sizes by accounting for unsuitable terrestrial environments, which we interpreted as proxies for global species ranges and calculated range centroids using QGIS (version 3.14.16)[98]. Then, we calculated the geodesic distances on a sphere (km) between the sampling sites with

associated abundance values to obtain the distance to the species' range centroid, using the WGS84 co-ordinate reference system. Abundance and distance values were $\log_{10}$-transformed prior to statistical analyses to account for scaling inconsistencies. Species with unresolved species level taxonomies ($N = 106$) as well as species with fewer than five observations were omitted from the analysis. We calculated Spearman Rank Correlation Coefficients ($r_s$) between $\log_{10}$(abundance) and $\log_{10}$(distance) values (Fig. 2). Negative $r_s$ values are consistent with an 'abundant-centre' distribution (Fig. 2).

## Spatial scale effects on abundance–distance relationships

We used three measurements to attempt to explore the effect of scale on 'abundant-centre' patterns: 1) we calculated the study extent (km), i.e., the spatial extent at which the study was conducted, encompassing the total study area between sampling locations in the data. This was measured using both latitudinal and longitudinal measurements. Initially, we used four categorical levels: 'Local' $\leq 250$ km, 'Landscape' > 251–500 km, 'Regional' > 501–1500 km and 'Continental' > 1501 km. 2) We calculated the grain ($km^2$), i.e., the spatial scale at which data were collected, which is important because the area of the base unit defines the spatial scale of the study[99], and variation in grain may be reflected in population abundance estimates[100]. Grain was extracted from each study by taking the base unit area for each sampling technique, e.g., sampling units measured in $km^2$. 3) We calculated study focus ($km^2$), defined as the spatial scale at which data were analysed. Often, abundance estimates from individual sampling sites are averaged across larger sampling areas, e.g. a protected area sampled using a number of line transects and multiple sampling points along each transect, with abundance values averaged across all the sampling points to produce an estimate for each transect. In most cases, grain and focus were the same for a given study, e.g. when abundance data were recorded in the form of points within a species' geographic range. Grain and focus values were $\log_{10}$ transformed due to the large variation in the range of these values. We then plotted the distribution of the $\log_{10}$-transformed values on separate histograms and visualised the natural breaks in the data. Using these we binned both grain and focus into two new categorical levels: 'small' and 'large' (−10 to −3 and −3 to 3 on $\log_{10}$ scale, respectively). We decided to drop extent from our analyses due to uneven sample sizes: data for only three groups (birds, mammals and plants) global 2717 species vs. local 146 species. We also dropped focus from our analyses because the natural break categorical bins were identical to those for grain. Grain was subsequently omitted from the statistical analyses due to its strong correlation with animal species group and plant functional group variables and thus could not be included within the same models due to colinearity.

## Compilation of dispersal-related species traits and geographic variables

We compiled six dispersal-related species traits for animal and eight traits for plant species to examine their effects on abundance–distance relationships (Table 1; Table S2). Traits were selected based on the morphological and/or ecological characteristics of the study species and included: for animals 1) taxonomic group (categorical), 2) body size (continuous), 3) invasive status (binary 1,0) and 4) feeding guild (categorical); and for plants: 1) functional group (categorical), 2) mean plant height (m), 3) seed mass (mg), 4) invasive status (binary 1,0), 5) life span (categorical) and 6) life form (categorical) (see Table 1 for an overview and Table S3 for justifications for the inclusion of each species trait/geographic variable). To explore spatial patterns within global abundance–distance relationships, we also compiled geographic data for species-level range sizes ($km^2$) and absolute latitudes (°).

To examine the effects of body size, we compiled body mass (g) data for mammals and birds, and snout-vent lengths (SVL; cm) for freshwater and reef fishes to produce the trait variable 'body size'. For plants, we used mean plant height (m) as a proxy for body size. Mean

plant height was used instead of maximum plant height as these were the only data available for our selected species, and notable effects of plant height on species abundance patterns would be reflected in either measurement. Where plant height and seed mass data were unavailable, we supplemented our data with gap-filled measurements[101,102] which were estimated using Bayesian Hierarchical Probabilistic Matrix Factorization[103]. Trait data for plant functional groups were sourced from the BiolFlor database[104] and the corresponding levels 'dwarf shrub' ($N = 11$ species) and 'subshrub' ($N = 4$ species) were merged into the level 'shrub' to produce four distinct categorical levels: 'grasses', 'herbs', 'shrubs' and 'trees'. Plant life-form data followed the classification of Raunkiær[105] but due to small sample sizes for 'chamaephytes' (woody plants with perennating buds borne close to the soil surface), we merged these with the 'phanerophytes' (woody perennial plants with buds at a distance from the surface, such as trees and shrubs). Invasive status was assessed using a binary approach (1 = invasive and 0 = non-invasive) according to the Invasive Species Specialist Group's Global Invasive Species Database (http://www.iucngisd.org/gisd/). For both animal and plant species, absolute latitude (°) was calculated as the absolute value of the range centroid. The following species traits/geographic variables were $\log_{10}$-transformed prior to analysis: body size (cm; g), mean plant height (m), seed mass (mg) and range size ($km^2$) to account for right-skew within the data. We tested for, but did not find, collinearity between continuous explanatory variables using a correlation threshold value $\geq 0.70$ in the *hmisc*[106] package with visualisations created in the *pheatmap* package[107] (Fig. S1).

## Statistics and reproducibility

All data analyses were performed R (version 4.0.5)[108]. No statistical method was used to predetermine sample size. Due to small sample sizes ($N = 9$ species), invertebrate abundance data were excluded from the statistical analyses. Coefficients from Spearman's rank correlation between species' abundances and range-centre distances were transformed to Fisher's Z scores (to achieve approximate normality), using the package *metafor*[109]. We analysed the animal and plant data separately because of different sets of trait combinations and the different sampling methods (e.g., abundances estimated from line transects for animals vs. vegetation plot-based cover estimates for plants). In all analyses, the influence of each Fisher's Z score was weighted by the natural logarithm of the sample size (to down-weight the influence of sparsely sampled species).

## Grand mean effects and linear modelling

We calculated the grand mean Fisher's Z scores of abundance–distance relationships as weighted averages of Fisher's Z values (using weighted intercept-only models), which we determined to be significant if the approximated 95% confidence intervals of the estimated intercept did not include zero.

To explore the effects of dispersal-related species traits and geographic variables on abundance–distance relationships, we expanded the weighted intercept-only models by including the species traits and geographic variables as fixed effects (hereafter 'predictors'). We determined the significance of these predictors using a stepwise model selection approach in which we started with a full model and (stepwise) removed non-significant predictors (based on $F$-tests on the explained variation between nested models) until we reached the most parsimonious models that included only statistically significant predictors. From these most parsimonious models, we calculated the marginal effects of each significant predictor using the *ggeffects* package[110].

## Interactions between predictor variables

To explore any potential interaction effects between the suggested predictors, we applied a multimodel inference approach[111]. Separately

for animal and plant data, we constructed all weighted linear effects models for any combination and two-way interactions between all tested predictors, accumulating 40,069 models for the animal dataset and 1450 models for the plant dataset. All competing models were ranked by the AIC[111], which favours the models that explain the highest amount of variation with the smallest number of fixed and interaction effects (using the dredge function of the *MuMIn* package[112]). From the emerging set of best models (those with a ΔAIC of <2 from the best model), we extracted all included fixed effects and interaction terms into two interaction models (separately for animal and plant data) from which we calculated the marginal interactions effects again with the *ggeffects* package.

### Testing for phylogenetic signals in abundance–distance relationships

To test for phylogenetic signal in our models and model residuals, we assembled phylogenies for a data subset consisting of 1645 birds (on 1547 nodes), 1151 freshwater and reef fishes (on 1031 nodes), and 201 mammals (on 184 nodes) from the Open Tree of Life (OTL)[113–115] using the *rotl* R package[116]. The plant phylogeny was obtained from the phylogenetic backbone of sPlot (version 3.0)[101,117] which is also based on the OTL with additional resolution and included 589 species on 552 nodes.

Separately for birds, fishes, mammals, and plant data, we compared the AIC values between three generalized least square models (estimated with maximum likelihood) that included a phylogenetic correlation matrix[118] with three different values of Pagel's λ, a robust index of phylogenetic signal in continuous traits[119]. A λ value of zero indicates no phylogenetic signal, whereas a value of one indicates full branch length separation, i.e. full phylogenetic signal. In addition to this, we used a further λ value calculated from the distribution of Fisher's Z values to reflect the observed phylogenetic evolution with the *phylosig* function from the *phytools* package[120]. In addition to the model comparison, we tested whether the residuals from our linear models were significantly predicted by species' phylogenetic relatedness. For this, we re-calculated the intercept-only model, the most parsimonious model (with fixed effects), and the interaction model (with important fixed and interaction terms) for the birds, fishes, mammals, and plant data set and tested the significance of the phylogenetic signal using Pagel's λ again using the *phylosig* function from the *phytools* package. See supplement.

### Reporting summary

Further information on research design is available in the Nature Portfolio Reporting Summary linked to this article.

## Data availability

The abundance–distance correlation coefficients, trait, and range data generated in this study have been deposited in the Dryad database [https://doi.org/10.5061/dryad.zgmsbccj2]. Original abundance data sources are outlined in Table S6 in the supplementary information. Source data are provided with this paper.

## Code availability

R code associated with this article is has been deposited in the Dryad database [https://doi.org/10.5061/dryad.zgmsbccj2].

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

## Acknowledgements

The authors would like to thank the following for their assistance with data collation and analyses, thought-provoking discussions and ideas: Jonathan Belmaker, Itai Granot, Luca Santini, Marten Winter, Erik Welk, Reinhard Klenke, Annegreet Veeken and Tom Reader. This project is generously supported by a NERC-funded ENVISION Doctoral Training Partnership PhD Studentship titled "*Abundance within species' ranges: understanding species' responses to environmental change*" (NE/S007423/1 2436183) (C.T.P., S. P. B., H.B., R.F. & F.S.).

## Author contributions

C.T.P., S.P.B., H.B., R.F. and F.S. conceptualised the study, C.T.P. & M.S. curated the original dataset, C.T.P., O.B., H.B., S.K., G.J. A.H., R.F. and F.S. developed the methodology, C.T.P., S.P.B., H.B., S.K., M.S., R.F. and F. S. conducted validation of the methodology and results, C.T.P., O.B., H.B., G.J.A.H and S.K. performed the formal analysis, C.T.P., O.B., S.K. and G.J.A.H. contributed to the visualisation, C.T.P. wrote the original draft, C.T.P., S.P.B., O.B., H.B., S.K., G.J.A.H., R.F., and F.S. contributed to writing review and editing, C.T.P. administered the project, H.B., S.P.B., R.F. and F.S. contributed to resource gathering and funding acquisition, H.B., S.P.B., R.F. and F.S. supervised the project.

## Competing interests

The authors declare no competing interests.
