## [Transparent Peer Review file · Nature Communications]

Plants with higher dispersal capabilities follow ‘abundant-centre’ distributions but such patterns remain rare in animals

Corresponding Author: Mr Connor Panter

Version 0:

Reviewer comments:

Reviewer #1

(Remarks to the Author)

This is a review of “Species abundances often conform to ‘abundant-centre’ patterns depending on dispersal capabilities”. The authors perform a meta-analysis of existing abundant-centre research, exploring variation among animal and plant taxa, as well as the role of trait values on resulting correlations between distance to geographic centroid and abundance. Overall, I thought the paper was interesting and well-written. I have a couple of high-level concerns, and then some minor points, which I go into more detail on below.

high-level concerns

The authors pull data from a few key studies that have disproportionately more species in them than others, and then proceed to find much of the same things that the previous authors found. I do think that there’s novelty here, but the authors could be more direct with the main findings. The novelty, in my mind, is the use of traits, and much less on the combination of data. That is, around 93% of the data for animals come from two studies whose goal was to evaluate abundant-center relationships (Table 1; Shalom et al 2020 and Dallas et al. 2017), with an even more pronounced bias in plants (all of the data come from two studies).

Relatedly, it seems important to note the inherent differences among taxa in how they were sampled, and how this may manifest as an abundant-center relationship. The Waldock paper (maybe same data as the Shalom paper?) for instance, using the ReefLife database, has a clearly constrained geographic space that could be occupied, and based abundance counts on point transects along the reef. It’s possible that taxa-level differences in support are really just taxa-level differences in how sampling was performed, right?

The central idea seems to be that species with traits related to good dispersal ability tended to show abundant-center relationships more often, but how well do these species adhere to abundant-center ideas? There are no figures of variance explained at a species-level or anything like that, but simply some effect sizes that are generally pretty close to 0 as a function of species group or some trait. If Figure S1 were placed in main text, it might be a bit more clear of a representation that abundant-center relationships actually tended to be quite rare, and just as often positive as negative.

(An interesting aside to this is that the Waldock paper claimed to have found support for abundant-center relationships in reef fish, but based on Figure S1, the authors actually find positive relationships. Did the Shalom paper use ReefLife data as well? I guess Shalom also argue that they found evidence for the abundant-center. The authors may wish to discuss some of these previous assertions given the data?)

So much of the work post-Martinez et al. 2013 has acknowledged the assumptions of the abundant center in geographic space (e.g., a perfectly spatially autocorrelated environmental axis which structures population dynamics), highlighting the need to explore abundant-center relationships in geographic _and_ niche space together (or some would argue just niche

space). A better discussion of assumptions and a stronger justification for not exploring these relationships in species climatic niche space seems warranted.

Finally, it seems relevant to discuss the implications of this. The authors find overwhelmingly little evidence for abundant-center relationships, but then assert that dispersal traits drive abundant-center relationships. This relates to the point above about Figure S1 and clearly showing these underlying values. The rank correlations do not really allow much in the way for variance explained, but it'd be interesting to highlight just how weak the relationships are, and then build the narrative around species differences from there. The main difference being that the authors phrase a lot of statements as 'this taxa or this trait showed more support for abundant-center relationships', but if the underlying relationships are incredibly weak and span both sides of the distribution of the bounded space from -1 to 1 (as in Figure S1), can we honestly say any group, regardless of trait/taxa, is showing support for abundant-center ideas? I guess 'more support' than the opposite of what folks think will be the case is 'more support'?

minor points

line 89: around here (if not elsewhere when discussing dispersal traits) it seems worth discussing the underlying assumption of equilibrium population dynamics across a species geographic range for abundant-center studies. That is, population fluctuations are trivial and the species is not actively expanding its geographic range (i.e., it would be a bit trivial to find an abundant-center for an actively spreading species, right?).

line 191: I might have missed it, but how do the authors account for species whose geographic range (if defined by a minimum convex polygon) overlaps portions of water? The analysis says it's global, and putatively combines data on a given species across studies (this was also a little unclear), such that it's possible that a species occurring in Europe and North America would have a really large portion of its geographic range in the ocean, right?

line 212: Dallas et al. 2017 used Spearman's as well (see Supplemental Figure S4), so the argument about the choice of statistical test being the cause for differences is not really valid.

On a higher-level note, if we have to choose really specific analyses (not necessarily rank correlations, which do make sense here) in order to get evidence for some pattern, have we really seen the pattern, or have we just used some math to convince ourselves of a preconceived notion? Not saying that the authors are doing this at all, but some parts of this paragraph feel like they miss an opportunity.

By focusing on differences among studies in statistical approaches, the authors ignore the inherent differences in sampling procedures etc. that could matter far more than how the geographic range is defined or what type of correlation was used. Finally, the bit around line 206 about the eBird data being flawed/unsuitable is a bit of a misnomer, right? The Fristoe study mainly argued that the speed of environmental change across the landscape made it such that we should not look for an abundant center, but just that abundance would be highest somewhere within the bounds of the species geographic range. But the data could still be used to look at abundant-center relationships in niche space, right? I just worry that calling one of the largest and most often used data resources in macroecology "unsuitable" might be a bit strong of a statement.

line 259: seems like this analysis would be fairly straightforward and would make for some good supplemental material.

Figure 2/3: this may be a bad/flawed idea, but I'm wondering how often significant effect sizes would be observed solely as a function of the underlying distributions of the data. That is, if the authors randomized a given variable and ran the model, would that always come out as non-significant? I guess I'm particularly interested in those variables that are inherently related (grasses, herbs, shrubs, and trees), as the paper spends time discussing these in the context of dispersal ability and group differences in abundant-center support.

Reviewer #2

(Remarks to the Author)

This paper tests the support for the geographic version of the abundant-centre hypothesis using data collected from several other studies. It found little support for the hypothesis, and - in line with several previous studies- found much variation around any detected central tendency. The study is well written and presented, although I have several important concerns regarding the goal and methods.

First, it is unclear to me why the authors decided to focus on the geographic version of the hypothesis. Since the paper by Martinez-Meyer et al. in 2013, most studies focused on the niche version of this hypothesis that has found more (yet still very unclear) support. The geographic version of the hypothesis has received even less support in the literature. Focusing on the geographic hypothesis puts emphasis on the assumption that species distributions are in equilibrium with the environment, but we know this to be untrue for many species, either because of dispersal limitations (e.g. Svenning and Skov 2004 *Ecol Lett*) or because of anthropogenic range loss (e.g. Britnell et al. 2023 *PNAS*) or recent expansion or range-shifts. Unsurprisingly, studies that tested both the geographic and niche centre hypothesis found very little consistency between the two patterns, and theoretical studies support the idea the two rarely overlap (Feng & Qiao 2022 *Glob Ecol Biogeog*). So overall, there are many evidence against the two conditions underlying the geographic version of the hypotheses (presented in lines 45-48). I think the author should first convince the readers that testing the geographic ACH is interesting and why we should expect such a pattern despite we know that those assumptions (L. 45-48) hardly hold true. I acknowledge that accounting for the niche while relying on many studies on different taxa may present several complications and limitations. However, the authors should at least try to relax those concerns, for example estimating the theoretical geographic centre (e.g. using historical ranges or niche-based predictions).

In addition to these points, I have some concerns on the methods. 1) I think absolute extent values make little sense, accounting for measures relative to the range size of the species would be more sensible; 2) The authors should explain why they chose each of the moderators, the role/expectation of some remains unclear (e.g. latitude), others would be hard to interpret unless combined (e.g. diet and size), and others may be missing (e.g. wing morphology, intent of the original study); 3) use of the entire range for migratory species; 4) The meta-analysis is not controlled for phylogenetic autocorrelation; 5) the use of concave polygons for marine species; 6) some clarifications needed on extent and grain; 7) a more in-depth discussion (and possibly consideration in the analysis) of biases in the literature. I elaborate on each of these points in my comments below.

Specific comments:

Title: The title seems to suggest species often conform to the hypothesis, however, the manuscript shows that animals did not adhere to this pattern, whereas plants did but not trees. Further, all taxa, irrespective of significant central tendencies, showed all possible relationships with ranging from positive to negative (Fig. S1), and except for invertebrates the median is always very close to zero. So overall, the title seems really misleading/unbalanced to me.

L. 7 and L26: Is it a goal for "population ecology" too? I'd say population ecology is mostly focused on investigating demographic mechanisms

L. 84-86: Another point is species space use in their daily activities (home range), the abundance of a species with a small a home range will vary in space more rapidly than a species with a large home range, so the small will less likely respond to macroecological variables and more likely respond to changes in microhabitat and microclimate.

L. 87-89: Ok, but this only consider climate, not habitat. Then, shall we also expect that species that are habitat generalist are more likely to stick to the rule?

L. 116-121: The authors should provide references that support the use of these traits as correlates for species dispersal abilities. However, I'm unconvinced using these traits is sufficient. For example, in mammals both body size and diet are known to be correlates of dispersal distance, but unless they are combined to predict dispersal using allometry, it is hard draw any conclusion on the individual effect of the two variables (e.g. a small carnivore can disperse more than a large herbivore). Also, several recent papers (e.g. Sheard et al. 2020 *Nat comm*, Weeks et al. 2022 *Fun Ecol*) have shown that wing morphology is an important driver of dispersal distance, more than dispersal as thought in the past (e.g. Sutherland et al. 2000 *Cons Ecol*).

L. 117: Why are you using latitude as a moderator? What do you expect? I couldn't find any justification for this.

L. 133: Since methods are at the end, briefly explain what factors were controlled for using random effects here.

L. 172-174: This is informative of the biases existing in the literature and casts doubts on our ability to answer this question with data collected this way. This reinforces my previous comment on the importance of differentiating studies that were meant to test the hypotheses, and studies that were not.

Furthermore, while the temporal trend is interesting, this test does not really tell us much for cherry picking. What if the studies not supporting the hypotheses were less likely to be published overall? This should be acknowledged and discussed.

L. 178-180: some of the ref cited here only focus on animals, and are in line with the results of this study (e.g. Hidas 2010, Dallas 2017, Santini 2018, Tuya 2008)

L. 191-192: Well, some studies were not bounded geographically either (e.g. Santini et al. 2018), but your study can still bear geographic biases present in the original studies. Furthermore, Fig. 1 shows that the data of most individual taxa have a strong geographic bias.

L. 192-218 this whole comparison with Dallas seems excessively long.

L. 236. Santini et al. predicted that dispersal was an important driver, but found no support to this hypothesis.

L. 314-315: what about species that have suffered range contraction? Note that range contraction is rarely unbiased and

often leads to marginalisation (Britnell et al. 2022, PNAS)

L. 314: What is a raw abundance value? Is it a simple count? Or is it an actual estimate of abundance? If counts, here you are missing the control for detectability, since at equal abundance the number of individuals detected can vary with the type of habitat.

If these are actual abundance estimates, are the study areas constant across sites? Or did you use abundance divided by study area (i.e. population density)?

L. 324-325: This can be seriously problematic, but at the same time not easily solvable. Since populations migrate, according to the ACH it is intuitive to expect abundance to decline from the centre to the edges of each of the two migratory ranges depending on the season. However, central populations in the breeding range do not necessarily arrive from the centre of the non-breeding range, the situation is generally quite messy with populations migratory routes crossing. Besides, migration is a process associated to high mortality, which probably correlates with the length of migratory route. So even within individual breeding and non-breeding ranges we may expect a lack of pattern. Overall, I suggest either removing migratory species, or only looking at the pattern in each seasonal range bearing in mind all due caveats.

L. 333. Using concave polygon is really unusual. I understand that a MCP may overlap with the land, but a less problematic approach would consist in simply masking the MCP by the sea area. Using convex polygons risks to decentralize substantially the range centre.

L. 338. Why are you long-transforming? transformation should not matter if you use Spearman correlation coefficients.

L. 345-348: Two problems here. You refer to extent but only indicate one linear measure (e.g. 250km). Is it the latitudinal or longitudinal extent? The geographic extent is given by two linear measures.

Second, I don't see why the absolute study extent should be important. Its effect is likely to be dependent on the range size of the species. If a species has a range that is smaller than 250km, then I'd not call it local, it is actually encompassing the whole range of the species.

L. 350: Please explain what you mean by "variation in grain may be reflected in population abundance estimates" and provide a supporting reference. I'm unclear if you refer to the effect of study area size on individual estimates, or the fact that studies conducted at different scale may provide inconsistent abundance estimates.

L. 400-405: Are you not controlling for phylogeny? Maybe not possible for invertebrates (?), but one could use trees truncated at higher taxonomic level than species, or taxonomic random effects), but it should be possible all other taxonomic groups tested.

L. 427: It is likely that in the literature there has been a serious cherry-picking problem, where positive results were more likely to be published than non-positive results. An important moderator that is missing is whether the data come from a study was meant to test the ACH or not. If the study that you used collected the data from the literature or existing databases, then it would be relevant to know if those data were collected for similar scopes or not.

Reviewer #3

(Remarks to the Author)

This contribution provides new perspectives on the abundant-centre hypothesis both by looking at a larger set of studies and by focusing on underlying traits that could explain the observations.

The study is important for both of these reasons, and I found no obvious issues with the study. The paper is very well written. The approach is clearly explained and both the introduction and conclusion sections are well written and do a good job of putting the study into context and explaining the significance of the results.

Version 1:

Reviewer comments:

Reviewer #2

(Remarks to the Author)

I thank the authors for addressing many of my comments, I think the paper has improved as a result. However, I am still not satisfied with how the hypothesis is presented, important methodological choices, and how the results are presented, interpreted and discussed. I'm also quite surprised that authors addressed some of my comments in the rebuttal letter only, without making any modification to the manuscript. This makes little sense to me, because the review process is not meant to convince the referees, but to make the paper bullet proof by addressing all concerns readers might come up with. If all the points raised below are adequately addressed (even though results and take-home message might change substantially), I believe the paper can be a good contribution to the literature.

I summarize my main comments below:

1) While I completely agree that testing the niche-version of the hypothesis presents many challenges, I'm also still very skeptical about the theoretical validity geographic version of the hypothesis. One point that must be made crystal clear in the

introduction and discussion, is that this hypothesis not only assumes species being in equilibrium with the environment, but also that environmental conditions influencing species abundance change gradually and monotonically from the centre to the edge of species range. To me, this is the main reason why it is hard to detect such pattern, and this becomes even more problematic for species with fine-scale habitat selection, where e.g. change in vegetation, orography, aspect, soil type, microclimate, availability of cover structures for nests, hiding, etc. etc. play a role in their abundance. We all know this is not true, at least not in all directions from the centre to the edge.

It also assumes all conspecific populations to be equally adapted to the environment, and recent research on SDMs has challenged this notion in some organisms, both small one e.g.

<https://www.nature.com/articles/s41598-020-79783-0>

OR <https://www.frontiersin.org/articles/10.3389/fpls.2019.01717/full>

as well as long-dispersing organisms

<https://www.pnas.org/doi/abs/10.1073/pnas.1820663116>

So, even though the average can show a little trend, a very large amount of noise is expected.

I'd like the authors to elaborate on this very clearly.

2) Fig. 2 clearly shows no pattern, however, the whole paper is written focusing on the species that do follow the pattern. I think the authors should emphasize more how rare this pattern is. What is the % of species that follows the pattern (while accounting for biases in species traits in the dataset)?

e.g. large vertebrates may be good at dispersing and perhaps follow the pattern, and maybe they make 50% of your dataset. But large vertebrates make a minority of vertebrate species.

Equally important, what percentage of the species that are expected to follow the pattern based on your results actually do it? e.g. how many long-distance dispersers actually show the expected pattern?

Finally, how many of the species that do not follow the pattern, do actually follow the inverse pattern?

I expect an honest discussion on the rarity of this pattern. This is important because it gives a sense of how likely results going in the expected direction are there by chance or for a reason.

Similarly, authors (as well as previous ones) do not attempt to explain why many species (apparently as many as those that meet the expectations) follow the reverse pattern. What is the mechanism underlying this? If significant inverse relationships are similarly frequent as those meeting expectations, how can we believe "good" results stem from the hypothesized mechanism?

3) Another serious problem I have with the manuscript is the interpretation of their results. They find that traits that are commonly used as dispersal proxies (body size, wing morphology, seed mass, etc...for which many supporting studies are available) are not significant, but taxonomic groups are. Based on this (and that large bodied mammals follow the pattern more often), they conclude dispersal is an important factor explaining differences (to the point this is highlighted in the title). Plus, the authors were quite dismissive on my comment on diet. My comment wasn't aimed at integrating diet in the test, but highlighting the simplification they were making. Take for instance the study of Sutherland et al. 2000 on dispersal distance of birds and mammals. According to their results, a carnivore bird of 500g disperses about 23.6 km, whereas an herbivorous/omnivorous bird of the same size disperses 1.8 km. Similarly, a carnivorous mammal of 1kg disperses 40.7 km, whereas herbivorous/omnivorous mammal of the same size disperses 3.3 km. So while body mass explains most of the variation in dispersal distance, it is not sufficient alone as a proxy of dispersal.

There is no need to add other proxies and interactions, the authors could use dispersal predictions based on allometric relationships where exist (for example, while reading the manuscript I wondered if the positive effect of body mass was consistent across diet types in mammals, this is very important because the proportion of herbivores and carnivores in the sample can alter your result).

Instead, wing morphology has been found as the only main predictor of bird dispersal distance (even though there's lot of unexplained variation), and yet, the authors did not find an effect. What I also don't understand, is why the authors tested it in response to my comment, but did not present it in the manuscript. This result is very relevant because it goes against your main conclusion that dispersal abilities underlies the abundant-centre pattern.

BTW, note that Sheard et al. now appears in the bibliography, but is not cited in the manuscript.

5) L. 164: The hypothesis on range size and latitude doesn't convince me. The reason why range size is larger at high latitude could be because of the more homogeneous conditions over large areas (longitudinally), and the more dominant effect of the environment over biotic interactions at higher latitudes. So interpreting an effect of range size as due to dispersal is risky.

Additionally, in the rebuttal the authors explained to me that latitude can account for Bergman and Allen's rule, which could affect species' dispersal abilities, but do not state it in the manuscript. In any case, this is highly speculative, I don't know a single study ever testing the effect of minor body mass or limb length modifications on species dispersal abilities, since all studies investigating dispersal drivers did it across species (not looking at intraspecific trait variation).

RESPONSE TO MY COMMENTS:

1) L. 559-563: The authors now added a bit in the discussion elaborating on my point on cherrypicking. But this point is not elaborated here in the methods where they introduce the publication bias analysis. Please explain exactly what the method tests for and what doesn't.

2) L 443. I don't understand authors' response on the need for log-transformation. They agree there is not need of log-transforming variables for spearman rank correlation, and yet they do it because to "account for the huge variation in

abundance/distance values between taxonomic groups" (?). Is it a visualization issue? Unclear, since all figures display using correlation coefficients. Please explain this better, and explain it in the manuscript (not to me).

3) I understand their response on "local/landscape/regional/continental", but then the difference should be made between studies that encompassed the whole range, vs studies that encompassed only part of it. Looking at the absolute range size is misleading in this sense. The expectation would be clear then, studies encompassing the whole range (irrespective of its absolute size) are more likely to detect the expected pattern.

4) Regarding my comment on phylogenetic control, the authors replied that: "convinced that it is appropriate to control for phylogeny in this study" but do not explain why, and that in a different study they are working on they "found no phylogenetic signals in abundance-distance slopes". However, again, this should be explained and presented in this paper, so that readers won't come up with my same question. Please elaborate on why you think this it is not appropriate OR present results showing the phylogenetic signal is absent (but then why you test it in a different study if you believe this to be inappropriate?).

I have to add, that while I'm not of the opinion phylogenetic control must always be considered, I believe you should in this case (see Cinar et al. 2022 *Met Ecol Evol*). Additionally, even if phylogenetic signal is low, many would argue you should still control for it. Note that considering phylogeny may reduce Error type 1, hence reducing the chance of significant results.

5) The authors acknowledged that recent range contraction can be a problem for their test, but just added a sentence in the discussion about it. I'm sorry but this is insufficient to me, recent range change can completely alter the results of this study, you cannot deal with this by simply acknowledging this might be a problem. The authors can and should test it, at least for those groups for which is possible. For some groups such as mammals, historical range for a subset of species is available (e.g. from Phylacine). For other groups the authors could check the RL assessments to see if the species is known to have contracted its range in recent times.

Minors:

L. 96-97: I don't think this is what the authors wanted to be their message. Weber et al. are fairly positive in presenting their results. Please clarify this is your conclusion.

Version 2:

Reviewer comments:

Reviewer #2

(Remarks to the Author)

I thank the authors for incorporating my comments, the presentation and discussion of the results is now more balanced. I only have a couple of remaining comments on the phylogeny.

L. 373-376: The authors compared the phylogenetically controlled with non-phylogenetically controlled models using AIC, and conclude the latter perform better. In all honesty, I don't know if this makes sense, I've never seen phylogenetic models being compared with non-phylogenetic models using information criteria. Clearly, accounting for phylogeny increases model complexity, but this is one of those cases where complexity might not need to be justified by model fit, but based on theory and expectations. If phylogenetic autocorrelation in the residuals is high, then phylogenetic control should be added irrespective of model complexity. Please provide a strong justification and supporting citations, or change it.

The authors also state "Our supplementary results showed that accounting for phylogeny reduced the significance of some moderator groups within our models indicating potential Type 1 Error"

This is expected, accounting for phylogeny will increase uncertainty and reduce the probability of detecting significant effects.

See e.g.

Chamberlain, S. A., et al. (2012). Does phylogeny matter? Assessing the impact of phylogenetic information in ecological meta-analysis. *Ecology Letters*, 15(6), 627-636.

Cinar, O., Nakagawa, S., & Viechtbauer, W. (2022). Phylogenetic multilevel meta-analysis: A simulation study on the importance of modelling the phylogeny. *Methods in Ecology and Evolution*, 13(2), 383-395.

L. 525-526: I don't understand the meaning of this sentence: "Abundance and distance values were log10-transformed, albeit this was not necessary, it 526 was performed in an attempt to rescale the variables prior to statistical analyses.". The authors admit it is not necessary, but they do it nonetheless to rescale the variables. First, rescaling should not influence spearman rank correlation. Second, if authors are interested in rescaling (why) should not use log-transformation, but can simply standardize to mean 0 and sd 1. Please clarify.

See also typo in L. 306: change "importer" to "importance" (?)

(Remarks on code availability)

Version 3:

Reviewer comments:

Reviewer #2

(Remarks to the Author)

I'm sorry I have remained the only referee to comment on this point and that I'm holding this paper back, but I disagree with virtually all responses given by the authors. I'll reply to each point raised by the authors:

“our approach does account for non-independence between species, as we weighted our models at the species-level which capture any species-level non-independence that may affect our modelled estimates....”

The only weights used in these models are those dependent on the sample size, which gives more relevance to correlations with large samples than those with small samples. This has nothing to do with independence. You also added a random effect for each effect size, again, this has nothing to do with phylogenetic non-independence.

“To date, there is no evidence within the literature of any effect of phylogeny on global abundance-distance relationships.....”

The fact that only one paper accounted for phylogenetic effects on the abundant-distant relationships doesn't provide any evidence that you don't need to control for it. Dallas et al. found no phylogenetic effect on a different dataset, and yet, presented only models corrected phylogenetically (in principle, if there is no phylogenetic signal, you should get comparable results). In your paper, you do not provide any evidence of lack of phylogenetic signal. You cannot cite unpublished data to support your argument, you need to present the evidence underlying your choices in the paper.

“we feel that our analyses (including the supplementary analyses) are transparent in regard to the effects of phylogeny on abundance-distance relationships....”

I disagree that the paper is transparent in this regard. The paper only present phylogenetically-controlled relationships in Table S12, and does not comment on them in the main text. Just reports: “increased uncertainty and reduced the significance of some moderator groups within our models, indicating potential Type 1 Error”. To be truly transparent, the authors need to elaborate on how repeating the analysis with phylogenetic correction changes their conclusions. Also, if the authors reject these alternative conclusions, need to justify why, and the justification cannot be “Dallas did not find a phylogenetic signal”, nor “AIC in the phylogenetically controlled model is higher”, as this is not how one determines whether phylogenetic correction is needed. The AIC evaluates model fit, but the phylogenetic control has nothing to do with improving model fit, but with the assumptions underlying the model. So the choice of a phylogenetic controlled vs non-phylogenetic controlled analysis depends on the research question (see e.g. Westoby et al. 2023 J Ecol) and dataset at hand. If you disagree, please provide supporting evidence that this is the right approach to determine whether phylogenetic control is needed.

In addition to the lack of discussion in the main text, I also find the discussion on this point in the supplementary materials highly unsatisfactory. The only interpretation given is:

“We suspect that the observed low phylogenetic effect may be due to the large number of genera and thus low number of species per genus included in our analysis”.

Personally, I don't understand how the number of genera should have an influence. Eventually, it depends on the phylogenetic signal. From a phylogenetic autocorrelation perspective, having many distant genera is less problematic than having many species in few genera.

I suggest the authors to discuss this point with an expert in phylogenetic comparative analyses.

In conclusion, I think that the authors need to:

- Be transparent on the phylogenetic signal in the data. If there is no phylogenetic signal, please report it and ignore all the rest, no need to bother further.
- If the phylogenetic signal is there, be transparent on how their conclusions change when controlling for phylogeny.
- They should also elaborate on why they trust more the non-phylogenetic results. How and why phylogenetic control is altering the results, and why the non-phylogenetic controlled model is more robust and trustable.

I want to stress that I am not imposing my own idea of how analyses should be conducted, I am well aware that there are several acceptable ways of conducting analyses and debates on methodological choices in our field are common. I'm just asking for an honest and transparent discussion of pros and cons of different approaches, especially because the phylogenetic-controlled approach is what is considered the golden standard in these sorts of analyses since several years now (Chamberlain et al. 2012 Ecol Lett). A transparent discussion can be useful for future papers.

(Remarks on code availability)

Reviewer #4

(Remarks to the Author)

The authors say they controlled for phylogeny and that their phylogenetic models had much larger AICs than non-phylogenetic models. This is presented in Table S12. However, I couldn't find where the authors spell out, either in R code or in the manuscript, how they did this. I also don't understand their reasoning for discarding the phylogenetic approach. One cannot escape phylogenetic assumptions when analysing cross-species data. If we run regressions (or any other statistical procedure) without phylogenetic effects then we are assuming a star phylogeny, ie that all species arose at the same time from their common ancestor. This has always seemed to be a very dubious assumption to me. I'm going to guess that the authors fitted phylogenetic models using the Brownian Motion assumption. If that is the case then high AIC values may just be an effect of the lack of validity of Brownian motion to describe the relationships. This is to be expected, especially on very large phylogenies where different evolutionary models might be expected to be more appropriate in different parts of the tree. An approach that might be a good compromise, and have even lower AIC values than those in Table S12, might be to use models that estimate Pagel's lambda statistic, which can provide a middle ground between full independence (a star) and Brownian motion. At the very least, the authors should look for phylogenetic signal in the residuals of their non-phylogenetic models to see if there is covariance there that should be modelled. Blomberg's K or Pagel's lambda can be used for this. I agree with the reviewer that complexity might not need to be justified by model fit, but based on theory and expectations.

Weighting the analyses by species does not control in any way for phylogenetic effects. The use of weighted least squares will account for heteroskedasticity in the data (some species may be more variable than others) but it doesn't address the covariance among species, which is what the authors (might, probably) need to model.

Part of the problem might be the authors' reliance on the metafor package to fit their mixed-effects models. Falling back to some better-tested and more flexible package, such as nlme would allow the authors to better examine the necessity for modelling phylogenetic covariance. Package ape, in conjunction with nlme can be used to fit Pagel's lambda models and a host of other possibilities.

Overall, I find the description of the methods with respect to the phylogenetic effects to be lacking, and the authors' rationale for not including phylogenetic effects to be disingenuous. It's impossible for me to judge the issue further.

The use of log transformations of the abundance and distance data: I agree with the reviewer that log transformation does not affect the Spearman's correlation coefficient at all, as it is a monotonic transformation that preserves the rank-order of the data, and the Spearman's is based only on the rank order. The advantage of log transforming the data is dubious unless it was important for other analyses. Often log transformation stabilises the variance, which may be helpful. Scaling to z-scores can help with model fitting too.

A minor point in the reporting of the statistics: In line 209 and 220, z statistics were presented, with degrees of freedom. But z is usually reserved for Normal tests, where there are no degrees of freedom. I think the authors may have meant t tests? The degrees of freedom are very large, so the t test should be equivalent to a z test, but it is good to maintain correct terminology.

(Remarks on code availability)

Version 4:

Reviewer comments:

Reviewer #2

(Remarks to the Author)

I thank the authors for taking the revisions seriously. I've read the revised version and I'm satisfied with the changes made that addresses my previous concerns. I'm confident this will be a highly influential paper on this much debated topic.

(Remarks on code availability)

The authors provide this private link for reviewers

<https://doi.org/10.5061/dryad.zgmsbccj2>

but the link seems to be outdated and is not accessible

Response to Reviewers Document: NCOMMS-23-12172-T

Reviewer #1 (Remarks to the Author):

Comment: This is a review of "Species abundances often conform to 'abundant-centre' patterns depending on dispersal capabilities". The authors perform a meta-analysis of existing abundant-centre research, exploring variation among animal and plant taxa, as well as the role of trait values on resulting correlations between distance to geographic centroid and abundance. Overall, I thought the paper was interesting and well-written. I have a couple of high-level concerns, and then some minor points, which I go into more detail on below.

>>> Thank you to the reviewer for their comments about our manuscript, please see below for point-to-point responses.

high-level concerns

Comment: The authors pull data from a few key studies that have disproportionately more species in them than others, and then proceed to find much of the same things that the previous authors found. I do think that there's novelty here, but the authors could be more direct with the main findings. The novelty, in my mind, is the use of traits, and much less on the combination of data. That is, around 93% of the data for animals come from two studies whose goal was to evaluate abundant-center relationships (Table 1; Shalom et al 2020 and Dallas et al. 2017), with an even more pronounced bias in plants (all of the data come from two studies).

>>> We thank the reviewer for their assessment of our manuscript providing novel insight and the excitement they show regarding our trait analyses. Respectfully, we feel that the novelty of our study lies in the combination of data *and* the use of species traits to ascertain new insights into the processes that mediate abundant-centre patterns, as we simultaneously assess these patterns across taxonomic groups and geographic space. In response to the reviewer's comment here, we have emphasised the novelty of using species traits to improve models of abundant-centre patterns across geographic space:

"Here, we show that using species-level traits to account for dispersal improves models of abundant-centre patterns across geographic space." (L39-41).

"Accounting for dispersal-related species traits improved our models of abundance–distance relationships" (L258-259).

"The use of species traits to explore global abundance–distance relationships enabled strong signals to be detected within a comparatively large data set." (L298-299).

"In the context of the taxonomic groups included in our analysis, our use of dispersal-related species traits suggests that dispersal capabilities must be accounted for when testing macroecological hypotheses such as the abundant-centre hypothesis." (L313-316).

"Despite this, our study indicates that accounting for dispersal-related traits improves our ability to model abundant-centre patterns across geographic space" (L364-365).

Comment: Relatedly, it seems important to note the inherent differences among taxa in how they were sampled, and how this may manifest as an abundant-center relationship. The Waldock paper (maybe same data as the Shalom paper?) for instance, using the ReefLife

database, has a clearly constrained geographic space that could be occupied, and based abundance counts on point transects along the reef. It's possible that taxa-level differences in support are really just taxa-level differences in how sampling was performed, right?

>>> The reviewer is correct to acknowledge the challenge of accounting for sampling biases within the abundance data. Within the 14 studies included in this analysis, there are even more studies that produce the underlying data. However, it must be noted that, as with all macroecological syntheses, our aim is to find signal amongst the noise and accounting for variation within individual sampling strategies is one of the main limitations of our study. Subsequently, we have produced an additional supplementary analysis exploring the effects of underlying abundance data source. Furthermore, we have produced a new supplementary table (see Tables S7 and S8 in supplementary material) which lists all underlying abundance data sources, with references, by taxonomic group, and have discussed this challenge within the discussion section.

“Abundance–distance relationships calculated from eBird and the USDA Forest Inventory Data (both from Dallas et al. (2017)), may be influenced by the sampling approaches introduced by these data sets. However, these signals may simply be an artifact of the species included within each data set, as duplicated species data between studies were removed prior to analysis and abundance data from the study with the higher number of observations was retained.” (L322-327)

Comment: The central idea seems to be that species with traits related to good dispersal ability tended to show abundant-center relationships more often, but how well do these species adhere to abundant-center ideas? There are no figures of variance explained at a species-level or anything like that, but simply some effect sizes that are generally pretty close to 0 as a function of species group or some trait. If Figure S1 were placed in main text, it might be a bit more clear of a representation that abundant-center relationships actually tended to be quite rare, and just as often positive as negative.

>>> We agree with the reviewer that Figure S1 allows for variation around median abundance-distance correlations to be visualised, in response, we have revised this figure (removing invertebrate plot as this group was dropped prior to analysis) and moved it into the main text to maximise the clarity of our findings for the reader (see Figure 2).

Comment: (An interesting aside to this is that the Waldock paper claimed to have found support for abundant-center relationships in reef fish, but based on Figure S1, the authors actually find positive relationships. Did the Shalom paper use ReefLife data as well? I guess Shalom also argue that they found evidence for the abundant-center. The authors may wish to discuss some of these previous assertions given the data?)

>>> Similar to our findings, Shalom et al. (2020) found wide variation in conformation to abundant-centre patterns in reef fishes. They concluded that changes in reef fish abundance across range cores were very small. The authors used data from both the Reef Life database and the GASPARE data set. We have now included an overview of source data sets which provide the abundance data for the studies included in our meta-analysis. The difference in support for abundant-centre relationships between Waldock et al. and the Shalom et al. (2020) paper may be due to the underlying data set, i.e., Shalom et al. (2020) also included data from GASPARE. In response to another reviewer's comments, we have explored the effects of the underlying data sets by running a series of additional meta-

regression models. The underlying data source explained approximately 6% and 14% of variation within the animal and plant data sets, respectively. Bird abundance data sourced from eBird (Dallas et al. 2017) showed a significant negative effect on abundance-distance relationships, indicative of support for “abundant-centre” patterns. Furthermore, tree abundance data from the USDA Forest Inventory and Analysis Database (also from Dallas et al. 2017) showed a strong significant positive effect on abundance-distance relations, which suggests overall no conformation to an “abundant-centre” pattern in North American trees. However, there was no effect of the ReefLife and/or GASPAR data sets which suggests that these differences may be due to study design/scope and/or sampling differences as previously mentioned by the reviewer. We have discussed these findings in the Discussion (see Tables S7 and S8 in supplementary material, as well as the supplementary methods).

“Similar to other tests of the abundant-centre hypothesis, there are caveats with regard to sampling within our data set. One such limitation relates to the underlying abundance data source. We explored the effects of underlying data sources on abundance–distance relationships as part of our supplementary analyses. Underlying data source contributed 6% and 14% of variation within our animal and plant abundance–distance relationships, respectively (Table S7; Table S8). Abundance–distance relationships calculated from eBird and the USDA Forest Inventory Data (both from Dallas et al. (2017)), may be influenced by the sampling approaches introduced by these data sets. However, these signals may simply be an artifact of the species included within each data set, as duplicated species data between studies were removed prior to analysis and abundance data from the study with the higher number of observations was retained.” (L318-327)

Comment: So much of the work post-Martinez et al. 2013 has acknowledged the assumptions of the abundant center in geographic space (e.g., a perfectly spatially autocorrelated environmental axis which structures population dynamics), highlighting the need to explore abundant-center relationships in geographic _and_ niche space together (or some would argue just niche space). A better discussion of assumptions and a stronger justification for not exploring these relationships in species climatic niche space seems warranted.

>>> As the reviewer points out, since the work of Martínez-Meyer *et al.* (2013), focus has shifted away from geographic space towards testing the hypothesis in ecological niche space. However, it must be noted that no clear consensus has been reached within the scientific community as to which type of “space” is most appropriate to test this highly debated hypothesis. Within our data set, there are huge differences in terrestrial and marine species microhabitats. Global environmental data would not accurately capture the niche of the species (at the micro-scale), and our data is not sufficiently high resolution. As with all macroscale patterns, any results may be the artifact of data limitations and may not be of any ecological relevance. To test the hypothesis in ecological niche space, we would have to use the same environmental space for all taxonomic groups using the same predictors. Some climatic predictors may also be more meaningful for some species but not for others. Additionally, the focus of this manuscript is dispersal which is related to geographic space. Calculating environmental niches for such a large data set with wide taxonomic and spatial coverage is non-trivial. It would thus go beyond the scope of this paper. However, we appreciate the reviewer comments and consider testing these ideas as part of another study.

Existing evidence remains unconvincing from both geographic and ecological niche space, with no clear patterns in either. Most importantly, fundamental research opportunities remain outstanding in geographic space. One such avenue includes the use of species traits, and dispersal capabilities, to attempt to explain a sizeable proportion of variation in global abundance-distance relationships. Our approach contributes significant advances to our

understanding of these relationships. As such, we have made this argument explicitly clear in paragraph 6 of the introduction

“Since the foundational work of Martínez-Meyer et al. (2013), exploration of abundant-centre patterns has shifted towards ecological niche space. However, no clear consensus has been reached within the scientific community as to which type of “space” is most appropriate to test this highly debated hypothesis, with evidence thus far remaining unconvincing. Calculating abundance–distance relationships in ecological niche space requires high resolution data, to accurately capture the niche of the species at the micro-scale, which are often unavailable for most macroecological studies. In addition to this, the same environmental space would be required across all taxonomic groups using the same climatic predictors which may be more meaningful to some species but not for others. As such, fundamental research opportunities remain outstanding in geographic space, which can be explored with available large-scale species distribution data. One of these potential avenues relates to linking species’ ecological characteristics, i.e., species traits, such as dispersal capabilities (Feng & Qiao 2022), to the species’ sampled abundances allowing for a sophisticated examination of the abundant-centre hypothesis using real-world data” (L117-130)

Comment: Finally, it seems relevant to discuss the implications of this. The authors find overwhelmingly little evidence for abundant-center relationships, but then assert that dispersal traits drive abundant-center relationships. This relates to the point above about Figure S1 and clearly showing these underlying values. The rank correlations do not really allow much in the way for variance explained, but it'd be interesting to highlight just how weak the relationships are, and then build the narrative around species differences from there. The main difference being that the authors phrase a lot of statements as 'this taxa or this trait showed more support for abundant-center relationships', but if the underlying relationships are incredibly weak and span both sides of the distribution of the bounded space from -1 to 1 (as in Figure S1), can we honestly say any group, regardless of trait/taxa, is showing support for abundant-center ideas? I guess 'more support' than the opposite of what folks think will be the case is 'more support'?

>>> In line with the reviewer’s comments, we have placed more emphasis on the fact that our abundance–distance correlations were, overall, relatively weak, i.e., with coefficients bounding either side of zero. Furthermore, we have included a new figure which clearly shows variation around median abundance-distance correlations, providing increased clarity of our findings to the reader (see Figure 2). Please see the following lines for in-text revisions relating to this point:

“Abundance–distance relationships depended on the taxa studied and were non-significant when summarised across all animal species and inconclusive when summarised across all plant species” (L32-35).

“Overall, abundant-centre patterns are not a general ecological phenomenon but can manifest in certain species with higher dispersal capabilities” (L39-39).

“Overall, global abundance–distance relationships were inconclusive. While animals showed no clear abundant-centre patterns (Fig. 2; Fig. 3a; $z = 1.501$, $df = 3,059$, $P = 0.133$), plants generally showed support for the abundant-centre hypothesis, i.e., a decrease in abundance with distance from their range centroids (Fig. 2; Fig. 3b; $z = -16.547$, $df = 614$, $P < 0.0001$).” (L197-200).

“Global abundance–distance relationships were relatively weak, with the distribution of abundance–distance correlations bounding both positive and negative values. Our findings are similar to previous research which yielded little to no support for the abundant-centre hypothesis (Samis & Eckert 2007; Tuya et al. 2008; Hidas et al. 2010; Dixon et al. 2013; Dallas et al. 2017; Santini et al. 2018; Ntuli et al. 2020; Sporbert et al. 2020). Despite this, some plant groups appeared to follow abundant-centre patterns whereas animals did not show any strong tendency toward ‘abundant-centre’ patterns.” (L252-258).

Figure 2.

minor points

Comment: line 89: around here (if not elsewhere when discussing dispersal traits) it seems worth discussing the underlying assumption of equilibrational population dynamics across a species geographic range for abundant-center studies. That is, population fluctuations are trivial and the species is not actively expanding its geographic range (i.e., it would be a bit trivial to find an abundant-center for an actively spreading species, right?).

>>> This is an interesting point which we have now included in a relevant point in the introduction. A test of the abundant-centre hypothesis would indeed consider population fluctuations as trivial and assumes almost a static equilibrium:

“Given this, the underlying assumptions of equilibrational population dynamics across a species geographic range assumes that population fluctuations are trivial, and that the species is not actively expanding its range” (L114-116).

Comment: line 191: I might have missed it, but how do the authors account for species whose geographic range (if defined by a minimum convex polygon) overlaps portions of water? The analysis says it's global, and putatively combines data on a given species across studies (this was also a little unclear), such that it's possible that a species occurring in Europe and North America would have a really large portion of its geographic range in the ocean, right?

>>> The reviewer raises an important point here, however, the majority of our geographic ranges were sourced from the expert derived range maps published on the IUCN Red List. Where these expert range maps were unavailable, we did use MCPs to infer geographic range, but this was only performed for small-ranged terrestrial species that did not overlap portions of water, i.e., expert range maps were usually unavailable for poorly-sampled/understudied species which are unlikely to have trans-continental ranges. This limitation has now been discussed in the Discussion section:

“We decided not to use the range size variable calculated by Dallas et al. (2017), who interpreted minimum convex polygons (MCPs) around sampling points as proxies for a species' range, and instead sourced most of our range size estimates from published IUCN expert range maps (IUCN 2022). Where expert range maps were unavailable, we used MCPs for some terrestrial species with small ranges, i.e., understudied species. It must be noted that MCPs have a coarse outer edge at low sampling intensities and are therefore sensitive to outliers (Burgman and Fox 2003), and sampling efforts only cover most of a species' range in exceptional circumstances (such as thorough field surveys of endemic species on small islands; Phiri et al. 2015).” (L275-283).

Comment: line 212: Dallas et al. 2017 used Spearman's as well (see Supplemental Figure S4), so the argument about the choice of statistical test being the cause for differences is not really valid.

>>> This argument has now been omitted from the revised manuscript. However, the point that we were trying to make here relates to the use of more robust analytical approaches, such as meta-regression models, when testing broad-scale relationships between species abundances and the environment. These analytical approaches are arguably more appropriate compared to simply analysing distributions of abundance-distance correlation coefficients (as seen in Dallas et al. 2017) when exploring global abundance-distance relationships. As such, we have emphasised this point in the Discussion:

“An explanation for this may simply lie in the number of dispersal-related traits used to test the hypothesis or the choice of methods used, i.e., more robust analytical techniques such as meta-analytical approaches, which may be more appropriate when testing broad-scale relationships between species abundances and their environments (see VanDerWal *et al.* 2009; Weber *et al.* 2017; Santini *et al.* 2018).” (L293-297).

Comment: On a higher-level note, if we have to choose really specific analyses (not necessarily rank correlations, which do make sense here) in order to get evidence for some pattern, have we *really* seen the pattern, or have we just used some math to convince ourselves of a preconceived notion? Not saying that the authors are doing this at all, but some parts of this paragraph feel like they miss an opportunity.

>>> This is a really interesting point, which relates to the comment above and calls into question the specificity and/or detail obtained from different statistical and analytical approaches in ecology. Santini *et al.* (2018) *Ecography* (<https://doi.org/10.1111/ecog.04027>) published some interesting results exploring differences in measures used to test the abundant-centre hypothesis but stopped short of testing the effects of different statistical approaches on abundance-distance relationships. We are sure that the reviewer will agree with us that this would be an interesting future research avenue.

Comment: By focusing on differences among studies in statistical approaches, the authors ignore the inherent differences in sampling procedures etc. that could matter far more than how the geographic range is defined or what type of correlation was used. Finally, the bit around line 206 about the eBird data being flawed/unsuitable is a bit of a misnomer, right? The Fristoe study mainly argued that the speed of environmental change across the landscape made it such that we should not look for an abundant center, but just that abundance would be highest *somewhere* within the bounds of the species geographic range. But the data could still be used to look at abundant-center relationships in niche space, right? I just worry that calling one of the largest and most often used data resources in macroecology "unsuitable" might be a bit strong of a statement.

>>> We have now reworded this sentence and altered the language to: “Furthermore Dallas *et al.* (2017) lends to their use of eBird data to test range-wide abundant-centre patterns in North American birds, which may introduce challenges associated with sampling procedures when calculating abundance–distance slopes (Fristoe *et al.* 2022)”. Regarding the comment about testing abundant-centre patterns in niche space, it appears that the jury is out within the literature. For example, Weber *et al.* (2017) *Ecography* found that relationships between species abundance and environmental suitability were weak or absent and could even be reversed. For a robust assessment of the abundant-centre hypothesis in niche space, one would have to assume strong(er) relationships between abundance and environmental conditions (potentially even stronger than tests in geographic space, as these may be (partially) explained by dispersal).

Comment: line 259: seems like this analysis would be fairly straightforward and would make for some good supplemental material.

>>> Using a subset data set for birds, we explored the effects of migratory strategy on bird abundance–distance slopes. We extracted migration strategy categorical data from the AVONET Database for 1,638 bird species and ran a mixed effect meta-regression model with abundance–distance slope fitted as the response term and migration strategy fitted as an explanatory moderator.

On average, bird abundance–distance slopes were negative which would indicate an abundant-centre pattern. However, migration strategy had a non-significant effect on bird abundance–distance slopes ($Q_M = 1.314$, $P = 0.519$). Given these additional findings, we are confident that the effects of migration strategy, in the case of this study, would yield non-significant results and would not alter the narrative of the manuscript. For reference, we have now included a supplementary methods and results section which contains this additional analysis (see supplementary material).

Table S9. Model results exploring the effects of migration strategy on global abundance–distance slopes (Fishers’ z) for 1683 bird species. Models are characterized by the between-species variance (τ^2), unaccounted heterogeneity (Q) and percentage of variability (I^2). Significant effects highlighted in bold. N spp. = number of species, SE = standard error, CI = confidence interval and df = degrees of freedom.

Moderator/subgroup	Estimate \pm SE	z	P	95% CI Range	τ^2	Cochran's Q	I^2 (%)	df	P
Migratory strategy									
Intercept	-0.312 \pm 0.010	-3.252	0.001	(-0.050) - (-0.012)	0.028	$Q_E = 14343.42$	92.58	1680	< 0.0001
partially	-0.012 \pm 0.017	-0.729	0.466	(-0.045) - (0.021)		$Q_M = 1.314$		2	0.519
sedentary	-0.014 \pm 0.012	-1.118	0.264	(-0.038) - (0.010)					

“Our supplementary analyses on a data set of 1,683 bird species suggests that migratory status has no effect on global abundance–distance relationships ($Q_M = 1.314$, $P = 0.519$; see Table S9), however, this aspect warrants further attention by future studies” (L349-352).

Comment: Figure 2/3: this may be a bad/flawed idea, but I'm wondering how often significant effect sizes would be observed solely as a function of the underlying distributions of the data. That is, if the authors randomized a given variable and ran the model, would that always come out as non-significant? I guess I'm particularly interested in those variables that are inherently related (grasses, herbs, shrubs, and trees), as the paper spends time discussing these in the context of dispersal ability and group differences in abundant-center support.

>>> This is an important point by the reviewer. We ran tests for collinearity between two moderators prior to inclusion in the full meta-regression models (see outputs presented in Figure S2). If we can understand what the reviewer suggests here clearly, we understand that this is already accounted for the moderators are already permuted in the analysis, this is built in already in the metafor R package, i.e., the Omnibuds test is based on permutation.

However, to ensure that we have sufficiently answered the reviewer’s concern here, we ran a series of additional Variation Inflation Factor (VIF) analyses on our animal and plant data set to explore multicollinearity between *at least three* moderator variables. We used the vif() function in the ‘metafor’ R package. A vif value >10 would indicate multicollinearity.

For animals, there was no indication of multicollinearity between moderator variables, i.e., all vif values < 10 (see Table below). However, for plants we detected potential multicollinearity between the functional group, range size, life form and life span moderators (see values highlighted in bold below). Ideally, we would remove these from the models but doing so would significantly alter the scope of the study.

Outputs from the variation inflation factor (vif) analyses exploring multicollinearity between at least three moderators for 3,026 animal and 619 plant species.

Moderators/subgroup	coefs	m	vif	sif
Animals				
Group(reef fishes)	4	1	1.40	1.18
log ₁₀ Body size	6	1	1.36	1.16
Feeding guild(omnivore)	8	1	1.22	1.11
Group(mammals)	3	1	1.19	1.09
Feeding guild(herbivore)	7	1	1.17	1.08
Absolute latitude (°)	9	1	1.14	1.07
Invasive	5	1	1.10	1.05
Group(freshwater fishes)	2	1	1.09	1.05
log ₁₀ Range size (km ²)	10	1	1.06	1.03
Plants				
Functional group(herbs)	2	1	34.18	5.85
Functional group(trees)	4	1	32.98	5.74
log₁₀ Range size (km²)	15	1	20.09	4.48
Life form(hemicryptophytes)	6	1	14.88	3.86
Life form(phanerophytes)	7	1	12.76	3.57
Life span(perennials)	12	1	10.08	3.17
Life form(therophytes)	8	1	9.34	3.06
Functional group(grasses)	1	1	7.17	2.68
log ₁₀ Mean plant height (m)	13	1	5.98	2.45
Functional group(shrubs)	3	1	4.59	2.14
Absolute latitude (°)	14	1	3.16	1.78
log ₁₀ Seed mass (mg)	16	1	2.21	1.49
Life span(biennials)	10	1	1.47	1.21
Life span(annuals/biennials)	9	1	1.22	1.10
Life span(biennials/perennials)	11	1	1.19	1.09
Invasive	5	1	1.11	1.05

Instead, we ran a simplified plant moderator model with potential confounding moderators removed (functional group, range size, life form and life span). Plant abundance–distance slopes were fitted as the response term and the remaining traits fitted as explanatory moderators. Again, this model was weighted by species sample size.

We also re-ran the full plant moderator model to compare the proportion of variance/heterogeneity explained by each. Both the simplified and full models were run with intercepts included to obtain an unadjusted R^2 value, using a maximum-likelihood (ML) method.

Both models were significant predictors of plant abundance–distance slopes (simplified: $Q_M = 92.9$, $P < 0.0001$; full: $Q_M = 622.9$, $P < 0.0001$). However, the simplified model explained 15.9% of variation within the data, whereas the full plant model explained 33.2% of variation.

This suggests that, despite the risk of multicollinearity between some moderators, the full plant model is better at explaining plant abundance–distance slopes.

Reviewer #2 (Remarks to the Author):

Comment: This paper tests the support for the geographic version of the abundant-centre hypothesis using data collected from several other studies. It found little support for the hypothesis, and - in line with several previous studies- found much variation around any detected central tendency. The study is well written and presented, although I have several important concerns regarding the goal and methods.

First, it is unclear to me why the authors decided to focus on the geographic version of the hypothesis. Since the paper by Martinez-Meyer et al. in 2013, most studies focused on the niche version of this hypothesis that has found more (yet still very unclear) support. The geographic version of the hypothesis has received even less support in the literature. Focusing on the geographic hypothesis puts emphasis on the assumption that species distributions are in equilibrium with the environment, but we know this to be untrue for many species, either because of dispersal limitations (e.g. Svenning and Skov 2004 *Ecol Lett*) or because of anthropogenic range loss (e.g. Britnell et al. 2023 *PNAS*) or recent expansion or range-shifts. Unsurprisingly, studies that tested both the geographic and niche centre hypothesis found very little consistency between the two patterns, and theoretical studies support the idea the two rarely overlap (Feng & Qiao 2022 *Glob Ecol Biogeog*). So overall, there are many evidence against the two conditions underlying the geographic version of the hypotheses (presented in lines 45-48). I think the author should first convince the readers that testing the geographic ACH is interesting and why we should expect such a pattern despite we know that those assumptions (L. 45-48) hardly hold true. I acknowledge that accounting for the niche while relying on many studies on different taxa may present several complications and limitations. However, the authors should at least try to relax those concerns, for example estimating the theoretical geographic centre (e.g. using historical ranges or niche-based predictions).

>>> We thank the reviewer for their thoughtful comments. We decided to focus on the geographic version of the hypothesis because species dispersal capabilities can only be meaningfully tested in geographic space. We argue that this would not be a useful approach to take when testing the abundant-centre hypothesis in niche space. Furthermore, there is an argument to be made that ecological niche space also assumes species distributions are in equilibrium with the environment, and there are many ways to measure a species' niche, e.g., species co-occurrences. Which method would best suite the abundant-centre hypothesis and how can we be sure that inferences made are not constrained by the approach taken and are ecologically meaningful? Given that current evidence for abundant-centre patterns from ecological niche space remains weak, it does not seem appropriate to extend the scope of this study further into ecological niche space.

Lack of consistency between detected patterns in geographic and ecological niche space further strengthens the approach to focus on one type of “space” and extend beyond what existing studies have done, e.g., in-depth exploration of the effects of species traits on global abundance–distance relationships as shown here. Assumptions of the ‘abundant-centre’ hypothesis are outlined in detail in the introduction (L105-112), i.e., 1) abundant-centre distributions represent a natural equilibrium of species populations in nature, 2) optimal

environments occur in the centre of a species' range, and 3) spatial autocorrelation of the environment (which cannot be achieved in ecological niche space).

As the reviewer rightly acknowledges, there are huge methodological and ecological constraints on testing the hypothesis in niche space, especially at such large taxonomic and spatial scales represented in our study. For example, global environmental data would not accurately capture the niche of the species (at the micro-scale), and our data is certainly not sufficiently high resolution. As with all macroscale patterns, any results may be the artifact of data limitations and may not be of any ecological relevance. To test the hypothesis in ecological niche space, we would have to use the same environmental space for all taxonomic groups using the same predictors. Some climatic predictors may also be more meaningful for some species but not for others. We have highlighted these challenges associated with testing the hypothesis in niche space, and have strengthened the argument for testing it in geographic space in the revised introduction:

“Since the foundational work of Martínez-Meyer et al. (2013), exploration of abundant-centre patterns has shifted towards ecological niche space. However, no clear consensus has been reached within the scientific community as to which type of “space” is most appropriate to test this highly debated hypothesis, with evidence thus far remaining unconvincing. Calculating abundance–distance relationships in ecological niche space requires high resolution data, to accurately capture the niche of the species at the micro-scale, which are often unavailable for most macroecological studies. In addition to this, the same environmental space would be required across all taxonomic groups using the same climatic predictors which may be more meaningful to some species but not for others. As such, fundamental research opportunities remain outstanding in geographic space, which can be explored with available large-scale species distribution data. One of these potential avenues relates to linking species’ ecological characteristics, i.e., species traits, such as dispersal capabilities (Feng & Qiao 2022), to the species’ sampled abundances allowing for a sophisticated examination of the abundant-centre hypothesis using real-world data” (L117-130).

Comment: In addition to these points, I have some concerns on the methods. 1) I think absolute extent values make little sense, accounting for measures relative to the range size of the species would be more sensible; 2) The authors should explain why they chose each of the moderators, the role/expectation of some remains unclear (e.g. latitude), others would be hard to interpret unless combined (e.g. diet and size), and others may be missing (e.g. wing morphology, intent of the original study); 3) use of the entire range for migratory species; 4) The meta-analysis is not controlled for phylogenetic autocorrelation; 5) the use of concave polygons for marine species; 6) some clarifications needed on extent and grain; 7) a more in-depth discussion (and possibly consideration in the analysis) of biases in the literature. I elaborate on each of these points in my comments below.

>>> We provide point-by-point responses to these issues below.

Comment: Title: The title seems to suggest species often conform to the hypothesis, however, the manuscript shows that animals did not adhere to this pattern, whereas plants did but not trees. Further, all taxa, irrespective of significant central tendencies, showed all possible relationships with ranging from positive to negative (Fig. S1), and except for invertebrates the median is always very close to zero. So overall, the title seems really misleading/unbalanced to me.

>>> Considering the tendency for abundance–distance slopes to bound either side of zero,

we have reworded the title to “Species abundances **can** conform to ‘abundant-centre’ patterns depending on dispersal capabilities”.

Comment: L. 7 and L26: Is it a goal for “population ecology” too? I’d say population ecology is mostly focused on investigating demographic mechanisms

>>> “Population ecology” now removed from the sentences in the abstract and introduction:

“Understanding patterns of biodiversity and the processes that drive them across varying spatial, temporal and taxonomic scales is a shared goal within macroecological and biogeographical research (Beck et al. 2012; Shade et al. 2018; McGill 2018).” (L45-46).

Comment: L. 84-86: Another point is species space use in their daily activities (home range), the abundance of a species with a small a home range will vary in space more rapidly than a species with a large home range, so the small will less likely respond to macroecological variables and more likely respond to changes in microhabitat and microclimate.

>>> This is a really interesting point, which is also largely dependent on the point in time (and space) when the abundance count was obtained by the original study. Because of the large scale of our study, and as the reviewer rightly points out, this would be very challenging to account for in this study due to species with smaller home ranges being unlikely to respond to our use of macroecological variables. We have added this point to the Discussion section:

“Our approach also did not account for species that have undergone range contractions which can often lead to ecological marginalisation (Britnell et al. 2023) and did not account for how species interact with the wider environment within their home ranges” (L329-332).

Comment: L. 87-89: Ok, but this only consider climate, not habitat. Then, shall we also expect that species that are habitat generalist are more likely to stick to the rule?

>>> Yes, the reviewer is correct to expect species that are habitat generalists may be more likely to follow abundant-centre patterns, as these species are better able to track changing environments. This expectation is dependent on the assumption that optimal environments occur in the centre of a species’ range which is one of the key assumptions of the abundant-centre hypothesis itself. We have now included this interesting point as a future research direction in the Discussion section:

“...and (iii) explore relationships between abundant-centre patterns and habitats at the macro- and micro-scales.” (L369-370).

Comment: L. 116-121: The authors should provide references that support the use of these traits as correlates for species dispersal abilities. However, I’m unconvinced using these traits is sufficient. For example, in mammals both body size and diet are known to be correlates of dispersal distance, but unless they are combined to predict dispersal using allometry, it is hard draw any conclusion on the individual effect of the two variables (e.g. a small carnivore can disperse more than a large herbivore). Also, several recent papers (e.g. Sheard et al. 2020 Nat comm, Weeks et al. 2022 Fun Ecol) have shown that wing morphology is an important driver of dispersal distance, more than dispersal as thought in the past (e.g. Sutherland et al. 2000 Cons Ecol).

Response:

We are not sure that using both body mass and feeding guild would yield any additional meaningful results here. Given that neither body mass nor feeding guild alone were significant individual predictors of abundance–distance relationships in mammals. Conversely, animal group (i.e., mammals, birds, reef fishes and freshwater fishes) did have significant individual effects, albeit positive. Multiple interaction models were run in an attempt to explore the effects of multiple species traits on abundance–distance relationships. Larger-bodied mammals were more likely to conform to abundant-centre patterns, as were mammals and freshwater fishes from higher latitudes. As such, whilst we agree with the reviewer that more comprehensive analyses of traits would be an interesting future research avenue, doing so within the scope of this study would result in loss of taxonomic resolution due to gaps within trait databases. We thus feel that our analyses are not only sufficient but also the most parsimonious choice given our comprehensive taxonomic coverage.

Trait justification

In response to the reviewer’s concern relating to trait justification, we have compiled the following table which clearly justifies the use of each species trait with a supporting reference and *a priori* expectations on global abundance–distance relationships.

Traits/moderators	Taxa	Justification of inclusion	Reference	A priori expectation
Morphological				
Group (categorical)	Animals	Dispersal capability differs between animal groups.	Bohonak (1999) https://doi.org/10.1086/392950	Groups that are better able to track changing environments, e.g., birds, will conform more to abundant-centre patterns. Larger-bodied animals will conform more to abundant-centre patterns as they are better able to track changing environments.
log ₁₀ Body size	Animals	Body size increases linearly with dispersal capability.	Stevens et al. (2014) https://doi.org/10.1111/ele.12303	Relative to smaller plants, taller plants are better dispersers and will conform more to abundant-centre patterns.
log ₁₀ Mean plant height (m)	Plants	Dispersal capability is correlated with plant height.	Thomson et al. (2011) https://doi.org/10.1111/j.1365-2745.2011.01867.x	
Ecological				
Functional group (categorical)	Plants	Dispersal capacity may be intrinsically linked with functional group, depending on the environment.	Aslan et al. (2019) https://doi.org/10.1093/aobpla/plz006	Faster growing and shorter-lived species, e.g., grasses and herbs, will conform more to abundant-centre patterns, as they are better able to track changing environments.
Invasiveness (1,0)	Both	Invasion fronts may benefit invasive species when dispersing.	Arim et al. (2006) https://doi.org/10.1073/pnas.0504272102	Invasive species will not conform to abundant-centre patterns, as abundant-edge distributions indicate invasion fronts.
Life span (categorical)	Plants	Shorter-lived species may be able to track changing environments more easily.	Beckman et al. (2018) https://doi.org/10.1111/1365-2745.12989	Shorter-lived plants, e.g., annuals and biennials, will conform more to abundant-centre patterns.
Life form (categorical)	Plants	Following Raunkjær’s system, the location of the growing bud may reflect a species’ dispersal ability as these relate directly to the environment.	Beckman et al. (2018) https://doi.org/10.1111/1365-2745.12989	Geophytes, therophytes, and hemicryptophytes will conform more to abundant-centre patterns, as they tend to be faster-growing and shorter-lived.

log ₁₀ Seed mass (mg)	Plants	Small-seeded species should disperse better than large-seeded species.	Venable & Brown (1988) https://www.jstor.org/stable/2461975	Plants with smaller seeds will conform more to abundant-centre patterns, as they are better able to track changing environments. Omnivorous species will conform more to abundant-centre patterns, as they are better able to track changing environments. Expected interaction effect between body size and feeding guild.
Feeding guild (categorical)	Animals	Feeding guild, particularly omnivores, influences dispersal capability.	Stevens et al. (2014) https://doi.org/10.1111/ele.12303	
Geographical				
log ₁₀ Range size (km ²)	Both	Dispersal capacity can be an important process moderating range size.	Lester et al. (2007) https://doi.org/10.1111/j.1461-0248.2007.01070.x	Species with larger ranges are better able to track changing environments will conform more to abundant-centre patterns.
Absolute latitude (°)	Both	Terrestrial species distributed at higher latitudes have larger ranges.	Ruggiero & Werenkraut (2007) https://doi.org/10.1111/j.1466-8238.2006.00303.x	Abundant-centre patterns should be more detectable in species that are distributed at higher latitudes.

Due to the constraints on the number of display items allowed, we have included this table in the supplementary material (Table S1) but have included our expectations towards the end of the revised Introduction in the main text:

“Based on previous research on animal distributions, we may expect to find differences between taxonomic groups in support for abundant-centre patterns due to variation in dispersal capabilities between taxa (Bohonak 1999), with larger-bodied animals conforming more to such patterns as they are better able to track changing environments (Stevens et al. 2014) (Table S1). Furthermore, an omnivorous diet has been shown to affect dispersal capability and thus we expect to find a dietary effect on abundance–distance relationships (Stevens et al. 2014). Dispersal capability in plants is known to correlate with height, and to a lesser degree with seed mass (Venables & Brown 1988), with taller species being able to disperse further together with those with smaller seeds (Thomson et al. 2011). Therefore, we expect to find effects of height and seed mass on plant abundance–distance relationships (Table S1). Similarly, species towards the faster end of the fast-slow life-history continuum may be better able to track changing environments. Consequently, we expect to find effects of functional group (Aslan et al. 2019), life span and life form (Beckman et al. 2018) on plant abundance–distance relationships (Table S1). More specifically, faster-growing species, e.g., grasses and herbs, shorter-lived species, e.g., annuals and biennials, should conform more to abundant-centre patterns relative to their longer-lived and slower-growing conspecifics, e.g., trees. In addition, if abundant-edge distributions accurately represent invasions fronts, i.e., benefiting invasive species when dispersing (Arim et al. 2006), then we expect invasive species to not conform to abundant-centre patterns (Table S1). Finally, dispersal capability can be an important process moderating range size (Lester et al. 2007), therefore, we would expect that species with larger ranges are better adaptors to the wider environment and thus will conform more to abundant-centre patterns. Similarly, terrestrial species distributed at higher latitudes tend to have larger ranges (Ruggiero & Werenkraut 2007), and we would therefore expect to find abundant-centre patterns being more detectable at higher latitudes (Table S1).” (L144-168).

Wing morphology

The reviewer also makes a very interesting point about wing morphology being a more suitable proxy for dispersal in birds. In response, we have conducted an additional analysis exploring the effects of wing morphology using the ‘Hand-Wing Index’ (HWI) variable in the AVONET Database.

We used a subset data set for 1,683 bird species and fitted the abundance–distance slopes as the response term and the HWI variable as the explanatory moderator. The mixed-effect meta-regression model was again weighted by bird species sample size to account for disproportionate sampling effort between species and studies.

Model results exploring the effects of wing morphology (using Hand-Wing Index as a proxy) on global abundance–distance slopes (Fishers’ z) for 1683 bird species. Models are characterized by the between-species variance (τ^2), unaccounted heterogeneity (Q) and percentage of variability (I²). Significant effects highlighted in bold. N spp. = number of species, SE = standard error, CI = confidence interval and df = degrees of freedom.

Moderator/subgroup	Estimate ± SE	z	P	95% CI Range	τ^2	Cochran's Q	I ² (%)	df	P
Hand-Wing Index (HWI)									
Intercept	-0.041 ± 0.012	-3.456	< 0.001	(-0.065) - (-0.018)	0.028	Q _E = 14409.24	92.62	1681	< 0.0001
HWI	< 0.001 ± < 0.001	0.1	0.920	(< -0.001) - (< -0.001)		Q _M = 0.010		1	0.920

We found no significant effect of wing morphology on abundance–distance relationships in birds (Q_M = 0.010, P = 0.92). Given these additional findings, we are confident that the inclusion of wing morphology would yield non-significant results and would not provide additional insights into dispersal-related drivers of global abundance–distance relationships in birds.

Comment: L. 117: Why are you using latitude as a moderator? What do you expect? I couldn't find any justification for this.

>>> Latitude was included as a moderator to control for any latitudinal effects specifically in relation to Bergmann's and Allen's Rules, (i.e., species at higher latitudes have larger body sizes and shorter limbs, respectively). These characteristics may affect a species' dispersal-capability. Assuming that a) species in our data set follow these rules and b) dispersal-capability has an effect on abundance–distance relationships, we expected to find a latitudinal effect on global abundance–distance relationships.

“Finally, dispersal capability can be an important process moderating range size (Lester et al. 2007), therefore, we would expect that species with larger ranges are better adaptors to the wider environment and thus will conform more to abundant-centre patterns. Similarly, terrestrial species distributed at higher latitudes tend to have larger ranges (Ruggiero & Werenkraut 2007), and we would therefore expect to find abundant-centre patterns being more detectable at higher latitudes (Table S1).” (L162-168).

In addition to this, a recent meta-analysis found widespread asymmetry in the performance of marginal (or range-edge) populations with profound latitudinal effects (see Pulido *et al.*

2023 *Global Ecology and Biogeography*). These patterns were largely driven by low-latitude marginal populations and suggests that latitude may be an important predictor of abundance–distance relationships in plants and animals.

Comment: L. 133: Since methods are at the end, briefly explain what factors were controlled for using random effects here.

>>> No random effects were included in the meta-regression models, instead we weighted each model by species sample size so that better-sampled species were assigned higher weights. This has been explained in the Methods section, and also now included at the beginning of the Results section:

“To account for differences in sampling intensity and effect size precision, we weighted each correlation coefficient by the number of the corresponding abundance observations.” (L194-196).

“We calculated the grand mean effect size of abundance–distance relationships with intercept-only models that included a random term for each effect size (a common meta-analytical practice to account for anticipated overdispersion in aggregated data).” (L523-525).

Comment: L. 172-174: This is informative of the biases existing in the literature and casts doubts on our ability to answer this question with data collected this way. This reinforces my previous comment on the importance of differentiating studies that were meant to test the hypotheses, and studies that were not. Furthermore, while the temporal trend is interesting, this test does not really tell us much for cherry picking. What if the studies not supporting the hypotheses were less likely to be published overall? This should be acknowledged and discussed.

>>> All of the 14 studies included in our analysis did test the abundant-centre hypothesis in some form, i.e., all were meant to test the hypothesis. However, the reviewer raises a very interesting point here relating to publication biases towards studies that find positive support for abundant-centre patterns. We have now included this point in the Discussion section:

“All of the studies included in our analyses specifically tested the abundant-centre hypothesis in some form. Relatedly, there may be a bias towards the publication of studies that find positive support for abundant-centre patterns over those that do not, which should be taken into consideration when interpreting our findings” (L356-359).

Comment: L. 178-180: some of the ref cited here only focus on animals, and are in line with the results of this study (e.g. Hidas 2010, Dallas 2017, Santini 2018, Tuya 2008)

>>> Another excellent point. We have now added Sporbert *et al.* (2020) which tested the abundant-centre hypothesis on European vascular plants and also Samis & Eckert (2007) which did not find strong support for abundant-centre patterns in two coastal dune plants:

*“...which yielded little to no support for the abundant-centre hypothesis (Samis & Eckert 2007; Tuya *et al.* 2008; Hidas *et al.* 2010; Dixon *et al.* 2013; Dallas *et al.* 2017; Santini *et al.* 2018; Ntuli *et al.* 2020; Sporbert *et al.* 2020).”* (L254-256).

Comment: L. 191-192: Well, some studies were not bounded geographically either (e.g. Santini et al. 2018), but your study can still bear geographic biases present in the original studies. Furthermore, Fig. 1 shows that the data of most individual taxa have a strong geographic bias.

>>> This is an important point, which we have now included in the Discussion section to maximise clarity for the reader:

“We employed a global approach to testing the abundant-centre hypothesis, however, our study is still bound by the geographical constraints of the studies included within our analyses. This is demonstrated by the clear geographical biases within our data set (see Figure 1) which may influence our findings.” (L337-341).

Comment: L. 192-218 this whole comparison with Dallas seems excessively long.

>>> We respectfully disagree with the reviewer on this. Until now, the study by Dallas et al. (2017) represented the largest test of the abundant-centre hypothesis in terms of taxonomic and spatial coverage. It would seem odd to not compare our findings with those of Dallas et al. (2017) in detail here. This is particularly important as a large proportion of our data derives from that study, and we therefore suggest the limitations associated with their approach should be clarified and outlined for the reader.

Comment: L. 236. Santini et al. predicted that dispersal was an important driver, but found no support to this hypothesis.

>>> Reference for Santini et al. (2018) removed from this sentence. The recent study by Feng & Qiao (2022) found that accounting for dispersal using simulated data improves understanding of species abundance patterns, therefore, we have revised this sentence accordingly:

“Our focus on species’ dispersal capabilities responded to recent research (Feng & Qiao 2022) which found dispersal to be an important process driving abundance–distance patterns within simulated environments.” (L307-309).

Comment: L. 314-315: what about species that have suffered range contraction? Note that range contraction is rarely unbiased and often leads to marginalisation (Britnell et al. 2022, PNAS)

>>> The reviewer raises an important point here relating to range contractions and ecological marginalisation. Unfortunately, our approach and data set does not allow for the inclusion of range contractions at the species-level, however, we have now included this important aspect as a discussion point along with an appropriate reference:

“Our approach also did not account for species that have undergone range contractions which can often lead to ecological marginalisation (Britnell et al. 2023) and did not account for how species interact with the wider environment within their home ranges” (L329-332).

Comment: L. 314: What is a raw abundance value? Is it a simple count? Or is it an actual estimate of abundance? If counts, here you are missing the control for detectability, since at equal abundance the number of individuals detected can vary with the type of habitat. If these are actual abundance estimates, are the study areas constant across sites? Or did you use abundance divided by study area (i.e. population density)?

>>> This is entirely dependent on the study rationale and approach taken by previous authors. Because we have used third-party data sets and performed a meta-analysis, across multiple taxonomic groups, accounting for differences in methodologies and sampling biases remains a key limitation of our study. We have acknowledged this and discussed potential implications in the revised Discussion:

“One such limitation relates to the underlying abundance data source. We explored the effects of underlying data sources on abundance–distance relationships as part of our supplementary analyses. Underlying data source contributed 6% and 14% of variation within our animal and plant abundance–distance relationships, respectively (Table S7; Table S8). Abundance–distance relationships calculated from eBird and the USDA Forest Inventory Data (both from Dallas et al. (2017)), may be influenced by the sampling approaches introduced by these data sets. However, these signals may simply be an artifact of the species included within each data set, as duplicated species data between studies were removed prior to analysis and abundance data from the study with the higher number of observations was retained” (L318-327).

Comment: L. 324-325: This can be seriously problematic, but at the same time not easily solvable. Since populations migrate, according to the ACH it is intuitive to expect abundance to decline from the centre to the edges of each of the two migratory ranges depending on the season. However, central populations in the breeding range do not necessarily arrive from the centre of the non-breeding range, the situation is generally quite messy with populations migratory routes crossing. Besides, migration is a process associated to high mortality, which probably correlates with the length of migratory route. So even within individual breeding and non-breeding ranges we may expect a lack of pattern. Overall, I suggest either removing migratory species, or only looking at the pattern in each seasonal range bearing in mind all due caveats.

>>> The reviewer raises some very interesting and important points here in relation to migratory species and temporal changes in abundance. Because the abundance data were sourced from other studies, we are unable to ascertain time stamps for when the abundance estimates were observed. Therefore, we are unable to separate the abundance data temporally/by seasonal ranges.

However, to address this complex issue of migration, we performed a supplementary analysis exploring the effects of migratory status on a subset data set for 1,683 bird species. We extracted migration strategy categorical data ('sedentary', 'migratory' and 'partially') from the AVONET Database for 1,638 bird species and ran a weighted mixed effect meta-regression model with abundance–distance slope fitted as the response term and migration strategy fitted as an explanatory moderator.

On average, bird abundance–distance slopes were negative which would indicate an abundant-centre pattern. However, migration strategy had a non-significant effect on bird abundance–distance slopes ($Q_M = 1.314$, $P = 0.519$). Given these additional findings, we are confident that the effects of migration strategy, in the case of this study, would yield non-significant results and would not alter the narrative of the manuscript. We have now included these additional analyses in the supplementary material (see Page 12-13 in the supplementary material).

Comment: L. 333. Using concave polygon is really unusual. I understand that a MCP may overlap with the land, but a less problematic approach would consist in simply masking the MCP by the sea area. Using convex polygons risks to decentralize substantially the range

centre.

>>> This approach was only performed on a handful of intercoastal bivalve species, which were not actually included in the analyses as we dropped data for invertebrates due to a lack of sufficient coverage. Furthermore, we only used the MCP method for terrestrial species that did not have global polygon range maps available. To avoid confusion and risks of miscalculating the range centroid highlighted by the reviewer, we have decided to remove the section relating to concave hulls in marine species from the methods.

Comment: L. 338. Why are you log-transforming? transformation should not matter if you use Spearman correlation coefficients.

>>> Yes, the reviewer is correct here. We are working with the spearman rank correlations therefore we do not expect log-transformation of the abundance and distance variables would have any effect on the results. Instead, our aim here was to account for the huge variation in abundance/distance values between taxonomic groups.

Comment: L. 345-348: Two problems here. You refer to extent but only indicate one linear measure (e.g. 250km). Is it the latitudinal or longitudinal extent? The geographic extent is given by two linear measures.

>>> Extent was measured using the two linear measurements indicated by the reviewer here, i.e., latitudinal and longitudinal. We have included this in the revised manuscript:

“This was measured using both latitudinal and longitudinal measurements” (L452).

Comment: Second, I don't see why the absolute study extent should be important. Its effect is likely to be dependent on the range size of the species. If a species has a range that is smaller than 250km, then I'd not call it local, it is actually encompassing the whole range of the species.

>>> Not necessarily, it depends on the scale of the study and sampling effort applied, i.e., larger-scale studies are likely to contain more data points per species and smaller-scale studies are unlikely to sample across the species' entire geographic range due to fieldwork/methodological/time constraints.

For example, Dallas et al. (2017) use data for some small-ranged rodent species but the study extent spans the entire North American continent. A small, local-scale study may only record abundance across smaller proportions of the species range and calculate abundance–distance slopes using these (e.g., Samis & Eckert 2007).

Our categorical study extent factor levels are dependent on the context and scale of studies included in our analysis, e.g., <250km is “local” relative to a global-scale study.

Comment: L. 350: Please explain what you mean by “variation in grain may be reflected in population abundance estimates” and provide a supporting reference. I'm unclear if you refer to the effect of study area size on individual estimates, or the fact that studies conducted at different scale may provide inconsistent abundance estimates.

>>> By “grain” we refer to the scale at which the abundance data were collected, e.g., 30 x 30m forest plots = 900m² (0.0009 km²) grain size (base unit area for each sampling technique). We attempted to capture the effects of changing grain size on species

abundance estimates by including grain size as a moderator variable in earlier models. The understanding behind this relates to weak but apparent relationships in abundance-occupancy models in relation to changes in grain size which were found by Caten *et al.* (2022) *Global Ecology and Biogeography* (see L457). The authors of that study point out that grain size effects on abundance-occupancy relationships remain widely inconsistent. Given this, we decided to control for this potential effect in our abundance–distance slopes but decided to remove the scale component of our analyses due to unequal sample sizes for each categorical factor level.

Comment: L. 400-405: Are you not controlling for phylogeny? Maybe not possible for invertebrates (? , but one could use trees truncated at higher taxonomic level than species, or taxonomic random effects), but it should be possible all other taxonomic groups tested.
>>> The exploration of phylogenetic effects is interesting, although we are not completely convinced that it is appropriate to control for phylogeny in this study. However, we are currently working on exploring phylogenetic signals across taxonomic levels in relation to abundance-distance relationships and other species traits (see Figure below) as part of another paper. For this, we tested for phylogenetic signals (using Blomberg's *K* and Lambda) at 1) the species-level, 2) within-genera and 3) within-families for birds, mammals and plants. At the species-level, which is the taxonomic level associated with the present study, we found no phylogenetic signals in abundance-distance slopes. As such, we are confident that it would not have any effects on the results of this paper.

Unpublished data. Phylogenetic signals (measured by Blomberg's K), in species traits across taxonomic levels for a) plants, b) mammals and c) birds. Phylogenetic signals were calculated across species, within-genera and within-families. Species-level data points represent absolute K values. Genus- and family-level boxplots show median K values with upper and lower interquartile ranges. Dashed grey line at $K = 1$ represents the threshold value for a detectable phylogenetic signal.

Comment: L. 427: It is likely that in the literature there has been a serious cherry-picking problem, where positive results were more likely to be published than non-positive results. An important moderator that is missing is whether the data come from a study was meant to test the ACH or not. If the study that you used collected the data from the literature or existing databases, then it would be relevant to know if those data were collected for similar scopes or not.

Response:

Scope of studies included in this analysis:

The reviewer raises an important point in relation to the publication of studies that find support for the ‘abundant-centre’ hypothesis, over studies that do not. All studies included in our analyses were originally designed to test the ‘abundant-centre’ hypothesis with variations in scale and taxonomic scope. Because of this, we are unable to include a new moderator variable within our meta-regression models on whether the study tested the ACH or not. We acknowledge that this was not made clear to the reader in the original manuscript, therefore, we have clarified this in the Discussion section as a study limitation:

“All of the studies included in our analyses specifically tested the abundant-centre hypothesis in some form. Relatedly, there may be a bias towards the publication of studies that find positive support for abundant-centre patterns over those that do not, which should be taken into consideration when interpreting our findings” (L356-359).

Effects of underlying data sources:

In response to the reviewer’s concerns regarding the effects of underlying data sources on observed abundance–distance relationships, we ran a series of supplementary analyses.

We used meta-regression models to explore the effects of underlying data sources on abundance-distance relationships. Models were only computed for data sources that contained data for ≥ 30 species. Within these models, “data source” was fitted as the explanatory moderator term, “abundance–distance correlation coefficient” was fitted as the response term and similarly to our full models, each model was weighted by species sample size to account for variations in sampling effort across taxa. We ran separate data source models for both animals and plants.

The underlying data source explained approximately 6% and 14% of variation within the animal and plant data sets, respectively. Bird abundance data sourced from eBird (Dallas *et al.* 2017) showed a significant negative effect on abundance-distance relationships, indicative of support for “abundant-centre” patterns. Furthermore, tree abundance data from the USDA Forest Inventory and Analysis Database (also from Dallas *et al.* 2017) showed a strong significant positive effect on abundance–distance relationships, which suggests overall no conformation to an “abundant-centre” pattern in North American trees. We have included these additional analyses within the supplementary material (Pages 10-11 in the supplementary material including Table S8) and have discussed the implications of the underlying data source in the discussion section:

*“One such limitation relates to the underlying abundance data source. We explored the effects of underlying data sources on abundance–distance relationships as part of our supplementary analyses. Underlying data source contributed 6% and 14% of variation within our animal and plant abundance–distance relationships, respectively (Table S7; Table S8). Abundance–distance relationships calculated from eBird and the USDA Forest Inventory Data (both from Dallas *et al.* (2017)), may be influenced by the sampling approaches introduced by these data sets. However, these signals may simply be an artifact of the*

species included within each data set, as duplicated species data between studies were removed prior to analysis and abundance data from the study with the higher number of observations was retained.” (L318-327).

Table S8. Model results exploring the effects of underlying data source on global abundance-distance slopes (Fishers’ z) for 3,049 animal and 614 plant species. Models are characterized by the between-species variance (z^2), unaccounted heterogeneity (Q) and percentage of variability (I^2). Significant effects highlighted in bold. N spp. = number of species, SE = standard error, CI = confidence interval and df = degrees of freedom.

Moderator/subgroup	N spp.	Estimate ± SE	z	P	95% CI Range	z^2	Cochran's Q	I^2 (%)	df	P	
Animals											
Intercept		0.031 ± 0.025	1.232	0.218	(-0.019) - (0.081)						
eBird	1559	-0.074 ± 0.026	-2.827	0.005	(-0.125) - (-0.023)						
EPA-EMAP and NAWQA	44	-0.057 ± 0.041	-1.385	0.166	(-0.137) - (0.024)	0.03	$Q_E = 70231.423$	97.19	3048	< 0.0001	
MCDB	113	0.010 ± 0.036	0.266	0.790	(-0.060) - (0.079)				$Q_M = 138.53$	5	< 0.0001
Reef Life Survey and GASPAR	1131	0.015 ± 0.026	0.575	0.565	(-0.036) - (0.066)						
TetraDENSITY	90	0.041 ± 0.037	-1.086	0.277	(-0.114) - (0.033)						
Plants											
Intercept		-0.100 ± 0.005	-19.803	< 0.0001	(-0.109) - (-0.090)	0.012	$Q_E = 22330.19$	97.75	613	< 0.0001	
USDA FIA	120	0.111 ± 0.013	8.760	< 0.0001	(0.086) - (0.136)				$Q_M = 76.73$	1	< 0.0001

Data source abbreviations: EPA-EMAP = The United States Environmental Protection Agency- Environmental Monitoring and Assessment Program; NAWQA = The National Water-Quality Assessment; MCDB = Mammal Community Database; GASPAR = General Approach to Species Abundance Relationships Database; USDA FIA = USDA Forest Inventory and Analysis Database.

Reviewer #3 (Remarks to the Author):

Comment: This contribution provides new perspectives on the abundant-centre hypothesis both by looking at a larger set of studies and by focusing on underlying traits that could explain the observations.

The study is important for both of these reasons, and I found no obvious issues with the study. The paper is very well written. The approach is clearly explained and both the introduction and conclusion sections are well written and do a good job of putting the study into context and explaining the significance of the results.

>>> Thank you to the reviewer for their kind and constructive comments about our study.

Reviewer #2 (Remarks to the Author):

I thank the authors for addressing many of my comments, I think the paper has improved as a result. However, I am still not satisfied with how the hypothesis is presented, important methodological choices, and how the results are presented, interpreted and discussed. I'm also quite surprised that authors addressed some of my comments in the rebuttal letter only, without making any modification to the manuscript. This makes little sense to me, because the review process is not meant to convince the referees, but to make the paper bullet proof by addressing all concerns readers might come up with.

If all the points raised below are adequately addressed (even though results and take-home message might change substantially), I believe the paper can be a good contribution to the literature.

I summarize my main comments below:

1) While I completely agree that testing the niche-version of the hypothesis presents many challenges, I'm also still very skeptical about the theoretical validity geographic version of the hypothesis. One point that must be made crystal clear in the introduction and discussion, is that this hypothesis not only assumes species being in equilibrium with the environment, but also that environmental conditions influencing species abundance change gradually and monotonically from the centre the the edge of species range. To me, this is the main reason why it is hard to detect such pattern, and this becomes even more problematic for species with fine-scale habitat selection, where e.g. change in vegetation, orography, aspect, soil type, microclimate, availability of cover structures for nests, hiding, etc. etc. play a role in their abundance. We all know this is not true, at least not in all directions from the centre to the edge.

It also assumes all conspecific populations to be equally adapted to the environment, and recent research on SDMs has challenged this notion in some organisms, both small one e.g.

<https://www.nature.com/articles/s41598-020-79783-0>

OR <https://www.frontiersin.org/articles/10.3389/fpls.2019.01717/full>

as well as long-dispersing organisms

<https://www.pnas.org/doi/abs/10.1073/pnas.1820663116>

So, even though the average can show a little trend, a very large amount of noise is expected.

I'd like the authors to elaborate on this very clearly.

>>> The reviewer raises a very important point here relating to a fundamental assumption of the abundant-centre hypothesis. We agree with the reviewer here and have incorporated all of their changes in relation to this point, paying particular attention to elaborating these assumptions very clearly in the introduction and discussion.

Introduction:

“The ‘abundant-centre’ hypothesis is built upon many assumptions

A major assumption of the hypothesis is that it assumes a species is in equilibrium with its environment and that environmental conditions influencing abundance change gradually and monotonically from the range centre towards the range edge; as implied by Grinnell (1922). Similarly, it also assumes that conspecific populations are equally adapted to their environments, with recent research using species distribution models challenging this notion in some species that demonstrate local adaptation across their range (Razgour et al. 2019; Chen et al. 2020; Goudarzi et al. 2021). It also assumes that optimal environments occur in the centre of a species’ range, that the environment is spatially autocorrelated. Accounting these broad assumptions, when testing the hypothesis...” (L106-115).

Discussion:

“Underlying assumptions may explain why global abundance–distance relationships remain relatively weak

A fundamental assumption of the abundant-centre hypothesis is that it assumes a species is in equilibrium with its environment and that environmental conditions influencing abundance change gradually and monotonically from the range centre towards the range edge. This assumption may explain why global abundance–distance relationships remained relatively weak across all taxonomic groups. Consequently, it may be even more challenging to detect abundant-centre patterns in species with fine-scale habitat selection, where changes in structural heterogeneity, orography, aspect, soil type, microclimate and availability of cover structures influence a species’ abundance across time and space.” (L293-302)

“Additionally, our test of the hypothesis assumes that conspecific populations are equally adapted to the environment. However, local adaptation across a species’ range persists for some species (Razgour et al. 2019; Chen et al. 2020; Goudarzi et al. 2021), which we were unable to account for using our approach” (L398-401).

2) Fig. 2 clearly shows no pattern, however, the whole paper is written focusing on the species that do follow the pattern. I think the authors should emphasize more how rare this pattern is. What is the % of species that follows the pattern (while accounting for biases in species traits in the dataset)?

e.g. large vertebrates may be good at dispersing and perhaps follow the pattern, and maybe they make 50% of your dataset. But large vertebrates make a minority of vertebrate species.

Equally important, what percentage of the species that are expected to follow the pattern based on your results actually do it? e.g. how many long-distance dispersers actually show the expected pattern?

Finally, how many of the species that do not follow the pattern, do actually follow the inverse pattern?

I expect an honest discussion on the rarity of this pattern. This is important because it gives a sense of how likely results going in the expected direction are there by chance or for a reason.

Similarly, authors (as well as previous ones) do not attempt to explain why many species (apparently as many as those that meet the expectations) follow the reverse pattern. What is the mechanism underlying this? If significant inverse relationships are similarly frequent as

those meeting expectations, how can we believe “good” results stem from the hypothesized mechanism?

>>> In response to the reviewer’s comments, we have now produced a table showing the outputs of the Spearman rank correlation coefficients between abundance and distance from the range centre for all species. Specifically, emphasis has been placed on providing an honest representation of all negative and positive correlation coefficients and the proportions of each which were statistically significant. In addition, we have included the percentage of species that bound either side of zero within the taxonomic groups presented in Figure 2. Finally, to clarify these underlying patterns for the reader, we have included the proportions of species that conform and do not conform to abundant-centre patterns in relevant positions within the results section:

“Abundance–distance relationships are relatively rare across taxonomic groups

Overall, global abundance–distance relationships were inconclusive. Animals showed no clear abundant-centre patterns (Fig. 2; Fig. 3a; Table 2; $z = 1.501$, $df = 3,059$, $P = 0.133$), with 47% ($n = 1,430$; $n_{total} = 3,060$) of animal species showing negative correlations between abundance and distance from the range centroid (indicative of an abundant-centre pattern). Of these negative correlations, 43% ($n = 620$; 1,430) were statistically significant, i.e., $P < 0.05$. Reef fishes and mammals generally did not conform to abundant-centre patterns, with 33% ($n = 370$; 1,131) and 50% ($n = 101$; 202) of species within these groups having negative abundance–distance correlations, respectively (Fig. 2e; Fig. 2g). Birds and freshwater fishes showed more support for abundant-centre patterns, however, these between-group differences were marginal with 56% ($n = 936$; 1,683) and 52% ($n = 23$; 44) of species showing negative abundance–distance correlations, respectively (Fig. 2a; Fig. 2c).

Plants generally showed support for the abundant-centre hypothesis, i.e., a decrease in abundance with distance from their range centroids (Fig. 2; Fig. 3b; $z = -16.547$, $df = 614$, $P < 0.0001$), with 75% ($n = 461$) of plant species having negative correlations between abundance and distance from the range centroid. Of these negative correlations, 75% ($n = 344$; 461) were statistically significant. Grasses, herbs and shrubs tended to follow abundant-centre patterns with 82% ($n = 53$; 65), 84% ($n = 337$; 401) and 62% ($n = 18$; 29) of species possessing negative abundance-distance correlations (Fig. 2b; Fig. 2d; Fig. 2f). Trees were the only plant subgroup not to follow broad abundant-centre patterns, with 44% ($n = 53$; 120) of species showing negative abundance-distance correlations (Fig. 2h).” (L206-226).

Table 2. Outputs from Spearman Rank Correlation Coefficients (r_s) exploring relationships between abundance and distance from geographic range centroids for 3,060 animal and 615 plant species. Negative r_s values, i.e., $r_s < 0$, indicate ‘abundant-centre’ patterns where species abundance is highest in the geographic range centre and declines towards the range edges. $r_s \geq 0$ indicates an inverse ‘abundant-edge’ pattern. Correlation coefficient weights in relation to the entire data set also presented. **Bold** values show abundance–distance correlations that exceed more than 50% of the moderator group. Percentage of statistically significant correlations ($P < 0.05$) also presented. Correlation coefficient summaries only shown for categorical moderators.

	N spp. (%/taxonomic group)	% total data set	$r_s < 0$ (%)	% $P < 0.05$	$r_s \geq 0$ (%)	% $P < 0.05$
Animals	3060 (100.0)	83.3	1430 (46.7)	43.4	1630 (53.3)	49.4
Group						
birds	1683 (55.0)	45.8	936 (55.6)	35.3	747 (44.4)	25.4
freshwater fishes	44 (1.4)	1.2	23 (52.3)	34.8	21 (47.7)	23.8
mammals	202 (6.6)	5.5	101 (50.0)	34.7	101 (50.0)	35.6
reef fishes	1131 (37.0)	30.8	370 (32.7)	21.8	761 (67.3)	75.6
Invasiveness						
Invasive	41 (1.3)	1.1	21 (51.2)	28.6	20 (48.8)	60.0
Non-invasive	3019 (98.7)	82.1	1409 (46.7)	43.6	1610 (53.3)	49.3
Feeding guild						
Carnivores	1436 (46.9)	39.1	619 (43.1)	44.9	817 (56.9)	53.1
Herbivores	610 (19.9)	16.6	288 (47.2)	42.0	322 (52.8)	49.1
Omnivores	1014 (33.1)	27.6	523 (51.6)	42.3	491 (48.4)	43.6
Plants	615 (100.0)	16.7	461 (75.0)	74.6	154 (25.0)	40.9
Functional group						
grasses	65 (10.6)	1.8	53 (81.5)	69.8	12 (18.5)	25.0
herbs	401 (65.2)	10.9	337 (84)	81.6	64 (16)	40.6
shrubs	29 (4.7)	0.8	18 (62.1)	77.8	11 (37.9)	63.6
trees	120 (19.5)	3.3	53 (44.2)	34.0	67 (55.8)	40.3
Invasiveness						
Invasive	23 (3.7)	0.6	20 (87.0)	95.0	3 (13)	0
Non-invasive	592 (96.3)	16.1	441 (74.5)	73.7	151 (25.5)	41.7
Life form						
Geophytes	20 (3.3)	0.5	16 (80.0)	81.3	4 (20.0)	25.0
Hemicryptophytes	256 (41.6)	7.0	218 (85.2)	80.3	38 (14.8)	26.3
Phanerophytes	186 (30.2)	5.1	99 (53.2)	52.5	87 (46.8)	43.7
Therophytes	153 (24.9)	4.2	128 (83.7)	81.3	25 (16.3)	56.0
Life span						
Annuals	66 (10.7)	1.8	51 (77.3)	78.4	15 (22.7)	46.7
Annuals/Biennials	10 (1.6)	0.3	9 (90.0)	88.9	1 (10.0)	0
Biennials	21 (3.4)	0.6	18 (85.7)	72.2	3 (14.3)	0
Biennials/Perennials	9 (1.5)	0.2	5 (55.6)	80.0	4 (44.4)	25.0
Perennials	509 (82.8)	13.9	378 (74.3)	73.8	131 (25.7)	42.0

3) Another serious problem I have with the manuscript is the interpretation of their results. They find that traits that are commonly used as dispersal proxies (body size, wing morphology, seed mass, etc...for which many supporting studies are available) are not significant, but taxonomic groups are. Based on this (and that large bodied mammals follow the pattern more often), they conclude dispersal is an important factor explaining differences (to the point this is highlighted in the title).

Plus, the authors were quite dismissive on my comment on diet. My comment wasn't aimed at integrating diet in the test, but highlighting the simplification they were making. Take for

instance the study of Sutherland et al. 2000 on dispersal distance of birds and mammals. According to their results, a carnivore bird of 500g disperses about 23.6 km, whereas an herbivorous/omnivorous bird of the same size disperses 1.8 km. Similarly, a carnivorous mammal of 1kg disperses 40.7 km, whereas herbivorous/omnivorous mammal of the same size disperses 3.3 km. So while body mass explains most of the variation in dispersal distance, it is not sufficient alone as a proxy of dispersal.

There is no need to add other proxies and interactions, the authors could use dispersal predictions based on allometric relationships where exist (for example, while reading the manuscript I wondered if the positive effect of body mass was consistent across diet types in mammals, this is very important because the proportion of herbivores and carnivores in the sample can alter your result).

Instead, wing morphology has been found as the only main predictor of bird dispersal distance (even though there's lot of unexplained variation), and yet, the authors did not find an effect. What I also don't understand, is why the authors tested it in response to my comment, but did not present it in the manuscript. This result is very relevant because it goes against your main conclusion that dispersal abilities underlies the abundant-centre pattern.

>>> We must apologise to the reviewer, as not including the wing morphology analysis in the revised manuscript appears to have been an oversight on our behalf during the previous round of revision.

We have now included this additional analysis in the supplementary material and also integrated these findings into the Discussion section:

“Wing morphology has been shown to be an importer predictor of dispersal distance in birds (Sheard *et al.* 2020; Weeks *et al.* 2022). To account for this, we conducted a supplemental analysis exploring the effects of the Hand-Wing Index (HWI) on global abundance–distance relationships for birds. For 1,683 species, we extracted HWI values from the AVONET database (Tobias *et al.* 2022). There was no significant effect of HWI on bird abundance–distance relationships (Table S10; $Q_M = 0.010$, $P = 0.92$), suggesting that dispersal capability in birds does not explain abundance–distance relationships in geographic space.” (L306-312).

We have carefully considered the reviewer’s comments relating to interpretation of the results, and agree that our “evidence” for dispersal-driven patterns in animals remains too weak. As such, we have made a concerted effort to emphasise that our original hypothesis relating to species with higher dispersal capabilities conform more to abundant-centre patterns does not hold true for animals. However, our data does appear to show that dispersal capability explains abundance–distance relationships in some plant groups, such as lower plants. Please see:

“Plants with higher dispersal capabilities follow ‘abundant-centre’ distributions but these patterns remain rare in animals” (Title; L1-2).

“Dispersal capability did not explain abundance–distance relationships in animals but likely explains such patterns in lower plants” (Abstract; L35-36).

“However, our findings suggest that dispersal capability does not explain abundance–distance relationships in animals but likely explains such patterns in lower plants. This suggests that species less limited by dispersal constraints are more likely to follow abundant-centre patterns in geographic space.” (Discussion; L289-291).

BTW, note that Sheard et al. now appears in the bibliography, but is not cited in the manuscript.

>>> Sheard *et al.* (2020) has now been cited in the manuscript (L307).

5) L. 164: The hypothesis on range size and latitude doesn't convince me. The reason why range size is larger at high latitude could be because of the more homogeneous conditions over large areas (longitudinally), and the more dominant effect of the environment over biotic interactions at higher latitudes. So interpreting an effect of range size as due to dispersal is risky.

Additionally, in the rebuttal the authors explained to me that latitude can account for Bergman and Allen's rule, which could affect species' dispersal abilities, but do not state it in the manuscript. In any case, this is highly speculative, I don't know a single study ever testing the effect of minor body mass or limb length modifications on species dispersal abilities, since all studies investigating dispersal drivers did it across species (not looking at intraspecific trait variation).

>>> Thank you to the reviewer for bringing this to our attention again. It appears that we were not as clear as we thought in our justification for including species range size and latitude as moderators/predictors of abundant-distance relationships – for which we apologise.

We agree with the reviewer in that range size and absolute latitude are not “species traits” and tend not to be used to explain a species' dispersal capability. Instead, we intended to include these moderators in our models to simply explore spatial patterns in abundance-distance relationships because our study is global in its extent. These moderators were not originally included within our list of “dispersal-related species traits”, however, we may have not been clear enough about this in previous versions of this manuscript. As such, we have revised certain areas of the manuscript to explicitly state that these moderators were included only to explore spatial patterns in our data set and are not/should not be interpreted as dispersal-related species traits:

Introduction

“To explore spatial patterns within global abundance–distance relationships, we also compiled geographic data for species-level range sizes (km²) and absolute latitudes (°).” (L185-186).

Methods

“Interactions between species traits and geographic variables

Separately within the animal and the plant dataset, we explored the predictive power of selected interactions between species traits and geographic variables” (L249-251).

“Compilation of dispersal-related species traits and geographic variables” (L559).

“To explore spatial patterns within global abundance–distance relationships, we also compiled geographic data for species-level range sizes (km²) and absolute latitudes (°).” (L568-570).

Due to speculation surrounding Bergman and Allen’s rules, we have decided to not include that argument in the manuscript.

RESPONSE TO MY COMMENTS:

1) L. 559-563: The authors now added a bit in the discussion elaborating on my point on cherrypicking. But this point is not elaborated here in the methods where they introduce the publication bias analysis. Please explain exactly what the method tests for and what doesn’t.

>>> Thank you for suggesting this, we have now revised this section and expanded on the what the methods (Eggers regression tests and funnel plots) test for:

“Funnel plots present effect sizes plotted against their standard errors or precisions (the inverse of standard errors) and visual examination of these can detect publication bias via the presence of a skew in the plot (Sterne *et al.* 2001). However, it is important to note that interpretation of these plots can be subjective. Therefore, to quantify publication bias we conducted a series of regression tests to explore correlations between effect sizes and sampling variance. Test outputs provide numerical correlation values, with strong correlations indicating potential publication biases (Lin & Chu 2018).” (L645-652).

2) L 443. I don’t understand authors’ response on the need for log-transformation. They agree there is not need of log-transforming variables for spearman rank correlation, and yet they do it because to “account for the huge variation in abundance/distance values between taxonomic groups” (?). Is it a visualization issue? Unclear, since all figures display using correlation coefficients. Please explain this better, and explain it in the manuscript (not to me).

>>> We did not log-transform the abundance and distance data for visualisation purposes but this was done in an attempt to rescale the variables (which differed greatly in their values) prior to statistical analyses. We appreciate that this is not necessary, as the reviewer highlights. However, we feel that the transformation of the raw abundance and distance data will not impact any of our results. We have attempted to clarify why this was done in the manuscript: “Abundance and distance values were log₁₀-transformed, albeit this was not necessary, it was performed in an attempt to rescale the variables prior to statistical analyses” (L525-526).

3) I understand their response on “local/landscape/regional/continental”, but then the difference should be made between studies that encompassed the whole range, vs studies that encompassed only part of it. Looking at the absolute range size is misleading in this sense. The expectation would be clear then, studies encompassing the whole range (irrespective of its absolute size) are more likely to detect the expected pattern.

>>> The reviewer raises an interesting point here, which we certainly agree with. However, due to methodological constraints associated with sampling across a species' entire range none of the studies included in our analyses actually encompassed the whole range of their study species. Therefore, albeit it is an interesting suggestion, we are unable to account for this in our results. We have included this as a Discussion point in the manuscript:

“Due to methodological constraints limiting estimates of abundance across a species' entire range, it must be noted that our abundance-distance relationships are based on abundance estimates from parts of a species' respective range.” (L419-422).

4) Regarding my comment on phylogenetic control, the authors replied that: “convinced that it is appropriate to control for phylogeny in this study” but do not explain why, and that in a different study they are working on they “found no phylogenetic signals in abundance-distance slopes”. However, again, this should be explained and presented in this paper, so that readers won't come up with my same question. Please elaborate on why you think this it is not appropriate OR present results showing the phylogenetic signal is absent (but then why you test it in a different study if you believe this to be inappropriate?).

I have to add, that while I'm not of the opinion phylogenetic control must always be considered, I believe you should in this case (see Cinar et al. 2022 Met Ecol Evol). Additionally, even if phylogenetic signal is low, many would argue you should still control for it. Note that considering phylogeny may reduce Error type 1, hence reducing the chance of significant results.

>>> Upon re-consideration we decided to test for potential phylogenetic bias in our analyses. To do this, we ran additional supplementary analyses for a subset of 1,645 birds, 1,151 fishes, 201 mammals and 589 plants. Assembling dated phylogenies from the Open Tree of Life (for animals) and sPlot 3.0 phylogenetic backbone (for plants), we computed additional grand mean meta-analytical random effects models that a) did not control for phylogeny and b) models that did. We then compared the model outputs for each taxonomic group by comparing differences in the Akaike Information Criterion, i.e., the goodness of fit. Our supplementary results showed that accounting for phylogeny generally yielded a higher AIC, i.e., a lower goodness of fit (increasing from 91 to 1,147 for birds, from -916 to 40 for fish, from 288 to 397 for mammals and from -783 to -335 for plants). We suspect that the observed low phylogenetic effect may be due to the large number of genera and thus low number of species per genus included in our analysis (which covered 697 bird, 347 fish, 109 mammal, and 286 plant genera).

Despite our supplementary analyses reducing the significance of some of our moderator groups, when comparing the AIC values, our models that did not control for phylogeny were better at explaining global abundance–distance relationships across these groups. To ensure that we are being as transparent for the reader as possible, we have included the outputs from our supplementary phylogeny analyses in the Supplementary Methods and discussed these in the Discussion section:

“Using a subset data set for 1,645 birds, 1,151 fishes, 201 mammals and 589 plants, we explored the effect of phylogeny on abundance–distance relationships (Table S12). Our supplementary results showed that accounting for phylogeny reduced the significance of some moderator groups within our models indicating potential Type 1 Error. However, the phylogeny-controlled models generally yielded higher Akaike Information Criterion values, suggesting that our models that did not control for phylogeny were better at explaining global abundance–distance relationships across these groups (Table S12; see Supplementary Material for more details).” (L369-376).

In addition to this, we have now also cited Cinar *et al.* (2021) in-text:

“...and phylogeny (Cinar *et al.* 2021).“ (L369).

5) The authors acknowledged that recent range contraction can be a problem for their test, but just added a sentence in the discussion about it. I'm sorry but this is insufficient to me, recent range change can completely alter the results of this study, you cannot deal with this by simply acknowledging this might be a problem. The authors can and should test it, at least for those groups for which is possible. For some groups such as mammals, historical range for a subset of species is available (e.g. from Phylacine). For other groups the authors could check the RL assessments to see if the species is known to have contracted its range in recent times.

>>> Upon reflection, we agree with the reviewer that we should have quantitatively tested the effects of historical range contractions on global abundance–distance relationships at least as part of the supplementary material. Due to data constraints, we tested for this using our data set for 202 mammal species that have available historical range data available. Using historical range data from the PHYLACINE database (version 1.2) (Faurby *et al.* 2018), we calculated the log ratio between the number of cells within a species' current range and divided this by the number of cells in the species' present natural range. Similar to our other supplementary analyses, we ran an additional mixed-effect meta-regression model, weighted by mammal species sample size, with the abundance–distance rank correlation coefficient fitted as the response term and the resulting log ratio range contraction variable fitted as an explanatory fixed effect term. There was a positive effect of historical range contraction on mammalian abundance–distance relationships, however, this effect was non-significant (Table S10; QM = 1.341, P = 0.247). As such, we have now revised the Discussion section regarding this matter and have provided an avenue for future research efforts.

Table S11. Model results exploring the effects of historical range contractions on global abundance–distance relationships (Fishers' z) for 202 mammal species. Historical range contractions measured by taking the log ratio of the number of cells in species' current range divided by the number of cells in the species' present natural range (see Faurby *et al.* 2018). Models are characterized by the between-species variance (τ^2), unaccounted heterogeneity (Q) and percentage of variability (I^2). Significant effects highlighted in bold. N spp. = number of species, SE = standard error, CI = confidence interval and df = degrees of freedom.

Moderator/subgroup	Estimate ± SE	z	P	95% CI Range	τ^2	Cochran's Q	I ² (%)	df	P
Mammals: Historical range change (HRC)									
Intercept	0.0309 ± 0.0341	0.9048	0.3656	(-0.0360) - (0.0978)	0.1576	QE = 5795.3130	93.86	200	< 0.0001
HRC	0.0876 ± 0.0757	1.1579	0.2469	(-0.0607) - (0.2359)		QM = 1.3408		1	0.2469

“Species that have undergone historical range contractions may become ecologically marginalised (Britnell et al. 2018), which may bias assessments of abundance–distance relationships. We attempted to account for these effects by exploring how abundance–distance relationships manifest in species that have undergone historical range contractions. Using historical range data from the PHYLACINE database (version 1.2; Faurby et al. 2018), we found no significant effect of historical range contractions on global abundance–distance relationships in mammals (Table S10; $Q_M = 1.341$, $P = 0.247$, $R^2 = 0.5\%$). Future research should leverage available historical range data across multiple taxonomic groups and explore, in detail, the effects of historical range dynamics on abundance–distance relationships.” (L402-411).

Minors:

L. 96-97: I don't think this is what the authors wanted to be their message. Weber et al. are fairly positive in presenting their results. Please clarify this is your conclusion.

>>> Thank you for this. We have now revised this sentence: “It is often assumed that species abundance is positively correlated with environmental suitability (an assumption of niche modelling), however, previous research has found that this is not always the case (see Weber *et al.* 2017). Therefore, this warrants further tests of abundance–distance relationships across geographic space” (L94-98).

REVIEWER COMMENTS

Reviewer #2 (Remarks to the Author):

I thank the authors for incorporating my comments, the presentation and discussion of the results is now more balanced. I only have a couple of remaining comments on the phylogeny.

>>> We would like to take this opportunity to thank the reviewer for committing a considerable amount of time to improve our manuscript. Since the initial submission the manuscript has improved substantially as a result of the reviewer's insightful suggestions.

L. 373-376: The authors compared the phylogenetically controlled with non-phylogenetically controlled models using AIC, and conclude the latter perform better. In all honesty, I don't know if this makes sense, I've never seen phylogenetic models being compared with non-phylogenetic models using information criteria. Clearly, accounting for phylogeny increases model complexity, but this is one of those cases where complexity might not need to be justified by model fit, but based on theory and expectations. If phylogenetic autocorrelation in the residuals is high, then phylogenetic control should be added irrespective of model complexity. Please provide a strong justification and supporting citations, or change it.

The authors also state "Our supplementary results showed that accounting for phylogeny reduced the significance of some moderator groups within our models indicating potential Type 1 Error"

This is expected, accounting for phylogeny will increase uncertainty and reduce the probability of detecting significant effects.

See e.g.

Chamberlain, S. A., *et al.* (2012). Does phylogeny matter? Assessing the impact of phylogenetic information in ecological meta-analysis. *Ecology Letters*, 15(6), 627-636.

Cinar, O., Nakagawa, S., & Viechtbauer, W. (2022). Phylogenetic multilevel meta-analysis: A simulation study on the importance of modelling the phylogeny. *Methods in Ecology and Evolution*, 13(2), 383-395.

>>> The reviewer raises an interesting point here relating to the inclusion of phylogenetic controls on our abundance-distance models. As the reviewer correctly pointed out, accounting for phylogeny will increase uncertainty in the models rendering some significant predictor variables non-significant (as highlighted by Chamberlain *et al.* 2012). Failure to account for phylogeny has been reported to violate independence assumptions within meta-regression models (Cinar *et al.* 2022).

We would like to emphasise that our approach does account for non-independence between species, as we weighted our models at the species-level which capture any species-level non-independence that may affect our modelled estimates. Therefore, our non-phylogenetically controlled models are robust in their ability to estimate global abundance-distance relationships.

There is a strong argument to make that accounting for phylogeny in modelled abundance-distance relationships is actually redundant anyway. To date, there is no evidence within the

literature of any effect of phylogeny on global abundance-distance relationships. The influential study by Dallas *et al.* (2017) *Ecology Letters* calculated abundance-distance relationships for 1,400 species and explored the effects of species phylogenetic relationships. Their study concluded that there was no association between abundance-distance patterns and phylogeny. Additionally, our unpublished analyses also found no species-level phylogenetic effects on modelled abundance-distance relationships for birds, mammals or plants adding to this pool of evidence (or lack of it). We did indeed find genera-level effects of phylogeny, however, this does not apply to the study in question as it was conducted at the species-level.

Despite this, in line with the reviewer's helpful comments, we feel that our analyses (including the supplementary analyses) are transparent in regard to the effects of phylogeny on abundance-distance relationships. Consequently, we respectfully disagree with the reviewer here and argue that our approach should include the non-phylogenetically controlled models with the phylogenetically controlled results presented in the supplementary information.

To further improve clarity for the reader, we have revised the text accordingly:

"Failure to account for phylogeny in meta-regression models may violate independence assumptions (Cinar *et al.* 2022). We weighted our meta-regression models by species to overcome this violation and additionally quantified the effect of phylogeny on our modelled abundance–distance relationships using a subset of our data set (1,645 birds, 1,151 fishes, 201 mammals and 589 plants (Table S12) to ensure robustness. Our supplementary results showed that accounting for phylogeny increased uncertainty and reduced the significance of some moderator groups within our models, indicating potential Type 1 Error (Chamberlain *et al.* 2012). However, evidence suggests that phylogeny has no effect on modelled abundance–distance relationships at the species-level (see Dallas *et al.* 2017; Panter *et al.* unpublished data), limiting any ecological rationale to account for it in species-level models within this context. In addition, our phylogeny-controlled models generally yielded higher Akaike Information Criterion values compared to those that did not account for phylogeny. Therefore, we conclude that models that did not control for phylogeny were better at explaining global abundance–distance relationships in this study (Table S12; see Supplementary Material for more details)." (L355-369).

L. 525-526: I don't understand the meaning of this sentence: "Abundance and distance values were log₁₀-transformed, albeit this was not necessary, it was performed in an attempt to rescale the variables prior to statistical analyses.". The authors admit is not necessary, but they do it nonetheless to rescale the variables. First, rescaling should not influence spearman rank correlation. Second, if authors are interested in rescaling (why) should not use log-transformation, but can simply standardize to mean 0 and sd 1. Please clarify.

>>> Thank you for highlighting this, we agree with the reviewer that this sentence in its current form is unclear. Therefore, we have attempted to improve the way it has been written to improve clarity and justify our use of log₁₀-transformations:

"Abundance and distance values were log₁₀-transformed prior to statistical analyses to account for difference in scale and improve comparability in abundance estimates between species, e.g., comparisons between large vs. smaller ranged species resulted in notable differences in distance from range centroid estimates." (L518-521).

See also typo in L. 306: change "importer" to "importance" (?)

>>> Thank you very much for spotting this, we have now corrected this (L307).

Reviewer #2 (Remarks to the Author):

I'm sorry I have remained the only referee to comment on this point and that I'm holding this paper back, but I disagree with virtually all responses given by the authors. I'll reply to each point raised by the authors:

“our approach does account for non-independence between species, as we weighted our models at the species-level which capture any species-level non-independence that may affect our modelled estimates....”

The only weights used in these models are those dependent on the sample size, which gives more relevance to correlations with large samples than those with small samples. This has nothing to do with independence. You also added a random effect for each effect size, again, this has nothing to do with phylogenetic non-independence.

“To date, there is no evidence within the literature of any effect of phylogeny on global abundance-distance relationships.....”

The fact that only one paper accounted for phylogenetic effects on the abundant-distant relationships doesn't provide any evidence that you don't need to control for it. Dallas et al. found no phylogenetic effect on a different dataset, and yet, presented only models corrected phylogenetically (in principle, if there is no phylogenetic signal, you should get comparable results). In your paper, you do not provide any evidence of lack of phylogenetic signal. You cannot cite unpublished data to support your argument, you need to present the evidence underlying your choices in the paper.

“we feel that our analyses (including the supplementary analyses) are transparent in regard to the effects of phylogeny on abundance-distance relationships....”

I disagree that the paper is transparent in this regard. The paper only present phylogenetically-controlled relationships in Table S12, and does not comment on them in the main text. Just reports: “increased uncertainty and reduced the

significance of some moderator groups within our models, indicating potential Type 1 Error". **To be truly transparent, the authors need to elaborate on how repeating the analysis with phylogenetic correction changes their conclusions.** Also, if the authors reject these alternative conclusions, need to justify why, and the justification cannot be "Dallas did not find a phylogenetic signal", nor "AIC in the phylogenetically controlled model is higher", as this is not how one determines whether phylogenetic correction is needed. The AIC evaluates model fit, but the phylogenetic control has nothing to do with improving model fit, but with the assumptions underlying the model. So the choice of a phylogenetic controlled vs non-phylogenetic controlled analysis depends on the research question (see e.g. Westoby et al. 2023 J Ecol) and dataset at hand. If you disagree, please provide supporting evidence that this is the right approach to determine whether phylogenetic control is needed.

In addition to the lack of discussion in the main text, I also find the discussion on this point in the supplementary materials highly unsatisfactory. The only interpretation given is:

"We suspect that the observed low phylogenetic effect may be due to the large number of genera and thus low number of species per genus included in our analysis".

Personally, I don't understand how the number of genera should have an influence. Eventually, it depends on the phylogenetic signal. From a phylogenetic autocorrelation perspective, having many distant genera is less problematic than having many species in few genera.

I suggest the authors to discuss this point with an expert in phylogenetic comparative analyses.

In conclusion, I think that the authors need to:

- Be transparent on the phylogenetic signal in the data. If there is no phylogenetic signal, please report it and ignore all the rest, no need to bother further.
- If the phylogenetic signal is there, be transparent on how their conclusions change when controlling for phylogeny.
- They should also elaborate on why they trust more the non-phylogenetic results. How and why phylogenetic control is altering the results, and why the non-phylogenetic controlled model is more robust and trustable.

I want to stress that I am not imposing my own idea of how analyses should be conducted, I am well aware that there are several acceptable ways of conducting analyses and debates on methodological choices in our field are common. I'm just asking for an honest and transparent discussion of pros and cons of different approaches, especially because the phylogenetic-controlled approach is what is considered the golden standard in these sorts of analyses since several years now (Chamberlain et al. 2012 Ecol Lett). A transparent discussion can be useful for future papers.

>>> We respect these counter-arguments and have completely revised all analyses to address the issue. This includes switching to weighted linear models instead of meta-regression models. We added robust tests for phylogenetic signals to our analyses.

Specifically, we used a phylogenetic correlation matrix to explore whether accounting for phylogenetic relatedness improves the grand mean models of abundance-distance relationships, using three values of Pagel's lambda (zero, one and lambda calculated from the distribution of Fisher's z-scores 'yi'), calculated with generalized least square regression models [Lines 644-665]. We found that controlling for phylogeny improved models of abundance-distance relationships for plants, but not for animals [Lines 287-294]. In addition, the plant grand mean effect became non-significant when phylogeny was controlled for suggesting that abundance-distance relationships in plants are partially influenced by phylogenetic relatedness [Lines 40-42; 289-291].

Although most of our conclusions were not much affected by the new analyses, those analyses have added some interesting new aspects to the results. Given these new results and in response to the reviewer's concerns regarding transparency of the effects of phylogeny on our models of abundance-distance relationships, we have added the following text in the sections indicated, including interpretation of phylogenetic signals in our models and model residuals:

Abstract

“Phylogeny improved models of abundance–distance patterns for plants but not for animals. Despite this, controlling for phylogeny yielded non-significant group-level results for plants, suggesting that only certain plant groups may conform to abundant-centre patterns. Overall, we demonstrate that abundant-centre patterns are not a general ecological phenomenon; they tend to not apply to animals but can manifest in certain plant groups, depending on dispersal capabilities and evolutionary histories“ (L39-44).

Introduction

“In addition, multi-species analyses could be biased due to phylogenetic non-independence between taxa (Gaston et al. 2007; Chamberlain et al. 2012). Lack of consideration and quantification of non-independence of species’ evolutionary relatedness limits our understanding of global abundance–distance relationships, resulting in calls for more robust studies incorporating appropriate comparative phylogenetic techniques (Martins & Hansen 1997). To date, other than Dallas et al. (2017), who explored the effects of body size and phylogenetic relatedness on support for the abundant-centre hypothesis, and Rivadeneira et al. (2010), who linked evolutionary history and traits to porcelain crab (Porcellanidae spp.) abundance distributions, there have been no examinations of the effects of species’ traits or phylogeny on support for the hypothesis across large taxonomic and spatial scales.

Here, we use a trait-based and comparative phylogenetic approach to conduct the largest test of the abundant-centre hypothesis to date, across five major taxonomic groups: birds, mammals, freshwater fishes, reef fishes and plants. Given that previous research has suggested that support for the hypothesis may be scale-dependent (Dallas et al. 2017; Santini et al. 2018) and abundance patterns might be closely related to species’ dispersal capabilities (see Santini et al. 2018; Dallas & Santini 2020; Feng & Qiao 2022), we also explore the effects of scale and body size.” (L141-157).

Results

Phylogenetic signals in abundance–distance relationships

Accounting for phylogenetic relatedness did not improve the model fit for the animal data set (Table 3). Conversely, for plants, incorporating the phylogenetic correction matrix with the observed value of Pagel's λ (~ 0.2) significantly improved the model fit and removed the significance of the estimated grand mean effect (Fisher's $Z = -0.04$, $t = -0.98$, $p = 0.33$). Similarly, our model residuals (tested separately for the bird, fish, mammal, and plant data sets) were mostly independent of species phylogenetic relatedness – again, except for the calculation of the grand mean effect in the plant data set (Table 3).” (L288-295).

Discussion

“Inclusion of phylogenetic relatedness in abundance–distance relationships returned a phylogenetic signal for certain taxonomic groups, indicating that abundance–distance relationships are not a general phenomenon but are restricted to phylogenetically clustered taxa.” (L314-317).

“Failure to account for phylogeny in ecological regression models may violate independence assumptions (Cinar et al. 2022), although previous evidence suggests that phylogeny has no broad effect on modelled abundance–distance relationships at the species level (Dallas et al. 2017). Here, we show for the first time that incorporating species' evolutionary histories in models of abundance–distance relationships can improve model robustness and generalizability. However, this effect appears to depend strongly on the taxonomic group studied, and was only significant in plant species for which common traits through shared ancestries might jointly influence the strength of the abundance–distance relationships. Pagel's λ assumes that traits follow a Brownian motion model of evolution (Revell et al. 2008) which is likely a broad oversimplification (but may also explain some of the observed patterns in our data set). Furthermore, it should be note that the metric used to calculate phylogenetic signal in our data set will likely influence the results obtained (see Münkemüller et al. 2012).” (L379-391).

Methods

“Testing for phylogenetic signal in abundance–distance relationships

To test for phylogenetic signal in our models and model residuals, we assembled phylogenies for a data subset consisting of 1,645 birds (on 1,547 nodes), 1,151

freshwater and reef fishes (on 1,031 nodes), and 201 mammals (on 184 nodes) from the Open Tree of Life (OTL; Redelings & Holder 2017, OpenTree et al. 2023a, 2023b) using the R package *rotl* (François et al. 2017). The plant phylogeny was obtained from the phylogenetic backbone of *sPlot* 3.0 (Bruehlheide et al. 2018; Hähn et al. 2024) which is also based on the OTL with additional resolution and included 589 species on 552 nodes.

Separately for birds, fishes, mammals, and plant data, we compared the AIC values between three generalized least square models (estimated with maximum likelihood) that included a phylogenetic correlation matrix (see Westoby et al. 2023) with three different values of Pagel's λ , a robust index of phylogenetic signal in continuous traits (Pagel 1999). A λ value of zero indicates no phylogenetic signal whereas a value of one indicates full branch length separation, i.e. full phylogenetic signal. In addition to this, we used a further λ value calculated from the distribution of Fisher's Z-values to reflect the observed phylogenetic evolution with the *phylosig* function from the *phytools* package (Revell 2024). In addition to the model comparison, we tested whether the residuals from our linear models were significantly predicted by species' phylogenetic relatedness. For this, we re-calculated the intercept-only model, the most parsimonious model (with fixed effects), and the interaction model (with important fixed and interaction terms) for the birds, fishes, mammals, and plant data set and tested the significance of the phylogenetic signal using Pagel's λ again using the *phylosig* function from the *phytools* package." (L641-662).

References

"Chamberlain, S. A., Hovick, S. M., Dibble, C. J., Rasmussen, N. L., Van Allen, B. G., Maitner, B. S. Ahern, J. R. Bell-Dereske, L. P., Roy, C. L., Meza-Lopez, M. *et al.* Does phylogeny matter? Assessing the impact of phylogenetic information in ecological meta-analysis. *Ecology Letters* **15**, 627-636 (2012)." (L756-759).

"Gaston, K. J., Chown, S. L. & Evans, K. L. Ecogeographical rules: elements of a synthesis. *Journal of Biogeography* **35**, 483-500 (2007)." (L813-814).

„Hähn, G. J. A., Damasceno, G., Alvarez-Davila, E., Aubin, I., Bauters, M., Bergmeier, E., Biurrun, I., Bjorkman, A. D., Bonari, G., Botta-Dukát, Z., *et al.* Global decoupling of functional and phylogenetic diversity in plant communities. *Nature*

Ecology & Evolution (2024) <https://doi.org/10.1038/s41559-024-02589-0>." (L830-833).

"Martins, E. P. & Hansen, T. F. Phylogenies and the comparative method: a general approach to incorporating phylogenetic information into the analysis of interspecific data. *The American Naturalist* **149**, 646-667 (1997)." (L897-898).

Münkemüller, T., Lavergne, S., Bzeznik, B., Dray, S., Jombart, T., Schiffrers, K. & Thuiller, W. How to measure and test phylogenetic signal. *Methods in Ecology and Evolution* **3**, 743-756 (2012). (L912-914).

"Pagel, M. Inferring the historical patterns of biological evolution. *Nature* **401** 877–884 (1999)." (L930).

Revell, L.J., Harmon, L.J. & Collar, D.C. Phylogenetic signal, evolutionary process, and rate. *Systematic Biology* **57**, 591-601 (2008). (L969).

"Westoby, M., Yates, L., Holland, B. & Halliwell, B. Phylogenetically conservative trait correlation: quantification and interpretation. *Journal of Ecology* **111**, 2105-2117 (2023)." (L1081-1083).

Reviewer #4 (Remarks to the Author):

The authors say they controlled for phylogeny and that their phylogenetic models had much larger AICs than non-phylogenetic models. This is presented in Table S12. However, I couldn't find where the authors spell out, either in R code or in the manuscript, how they did this. I also don't understand their reasoning for discarding the phylogenetic approach. One cannot escape phylogenetic assumptions when analysing cross-species data. If we run regressions (or any other statistical procedure) without phylogenetic effects then we are assuming a star phylogeny, ie that all species arose at the same time from their common ancestor. This has always seemed to be a very dubious assumption to me. I'm going to guess that the authors fitted phylogenetic models using the Brownian Motion assumption. If that is the case then high AIC values may just be an effect of the lack of validity of Brownian motion to describe the relationships. This is to be expected, especially on very large phylogenies where different evolutionary models might be expected to be more appropriate in different parts of the tree. An approach that might be a good

compromise, and have even lower AIC values than those in Table S12, might be to use models that estimate Pagel's lambda statistic, which can provide a middle ground between full independence (a star) and Brownian motion. At the very least, the authors should look for phylogenetic signal in the residuals of their non-phylogenetic models to see if there is covariance there that should be modelled. Blomberg's K or Pagel's lambda can be used for this. I agree with the reviewer that complexity might not need to be justified by model fit, but based on theory and expectations.

Weighting the analyses by species does not control in any way for phylogenetic effects. The use of weighted least squares will account for heteroskedasticity in the data (some species may be more variable than others) but it doesn't address the covariance among species, which is what the authors (might, probably) need to model.

Part of the problem might be the authors' reliance on the *metafor* package to fit their mixed-effects models. Falling back to some better-tested and more flexible package, such as *nlme* would allow the authors to better examine the necessity for modelling phylogenetic covariance. Package *ape*, in conjunction with *nlme* can be used to fit Pagel's lambda models and a host of other possibilities.

Overall, I find the description of the methods with respect to the phylogenetic effects to be lacking, and the authors' rationale for not including phylogenetic effects to be disingenuous. It's impossible for me to judge the issue further.

>>> Thank you for your comments and providing suggestions on how to account for phylogenetic assumptions when working with multispecies data sets, such as ours. We have taken your advice fully on board and have reworked our analyses in line with your comments. We believe these changes have much improved the robustness and indeed the relevance of our study.

For the main analysis, as suggested, we have now switched from the meta-analytical random-effects models provided in the *metafor* package to a stepwise selection approach for weighted linear effects models. In line with the reviewer's suggestion,

we made several adjustments. We now use t statistics-based tests to determine the significance of predictor and subgroup effects on Fisher's z values. We performed additional tests for phylogenetic signals in our coefficients and model residuals using three values of Pagel's lambda (zero, one and lambda calculated from the distribution of Fisher's z -scores) and generalized least square regression models with maximum likelihood. Our results now show that phylogenetic relationships were independent from the observed abundance-distance relationships for animals, but did improve model fit and robustness of abundance-distance relationships in plants, up to the point that the grand mean effect became non-significant with increasing model complexity for the plant data set. Given these new analyses, we added the following text:

“Phylogenetic signals in abundance–distance relationships

Accounting for phylogenetic relatedness did not improve the model fit for the animal data set (Table 3). Conversely, for plants, incorporating the phylogenetic correction matrix with the observed value of Pagel's λ (~ 0.2) significantly improved the model fit and removed the significance of the estimated grand mean effect (Fisher's $Z = -0.04$, $t = -0.98$, $p = 0.33$). Similarly, our model residuals (tested separately for the bird, fish, mammal, and plant data sets) were mostly independent of species phylogenetic relatedness – again, except for the calculation of the grand mean effect in the plant data set (Table 3).” (L288-295).

“Testing for phylogenetic signal in abundance–distance relationships

To test for phylogenetic signal in our models and model residuals, we assembled phylogenies for a data subset consisting of 1,645 birds (on 1,547 nodes), 1,151 freshwater and reef fishes (on 1,031 nodes), and 201 mammals (on 184 nodes) from the Open Tree of Life (OTL; Redelings & Holder 2017, OpenTree et al. 2023a, 2023b) using the R package `rotl` (François et al. 2017). The plant phylogeny was obtained from the phylogenetic backbone of `sPlot 3.0` (Bruehlheide et al. 2018; Hähn et al. 2024) which is also based on the OTL with additional resolution and included 589 species on 552 nodes.

Separately for birds, fishes, mammals, and plant data, we compared the AIC values between three generalized least square models (estimated with maximum likelihood) that included a phylogenetic correlation matrix (see Westoby et al. 2023) with three different values of Pagel's λ , a robust index of phylogenetic signal in continuous traits (Pagel 1999). A λ value of zero indicates no phylogenetic signal whereas a value of one indicates full branch length separation, i.e. full phylogenetic signal. In addition to this, we used a further λ value calculated from the distribution of Fisher's Z-values to reflect the observed phylogenetic evolution with the phylosig function from the phytools package (Revell 2024). In addition to the model comparison, we tested whether the residuals from our linear models were significantly predicted by species' phylogenetic relatedness. For this, we re-calculated the intercept-only model, the most parsimonious model (with fixed effects), and the interaction model (with important fixed and interaction terms) for the birds, fishes, mammals, and plant data set and tested the significance of the phylogenetic signal using Pagel's λ again using the phylosig function from the phytools package." (L641-662).

The use of log transformations of the abundance and distance data: I agree with the reviewer that log transformation does not affect the Spearman's correlation coefficient at all, as it is a monotonic transformation that preserves the rank-order of the data, and the Spearman's is based only on the rank order. The advantage of log transforming the data is dubious unless it was important for other analyses. Often log transformation stabilises the variance, which may be helpful. Scaling to z-scores can help with model fitting too.

>>> As described in the section 'Scale effects on abundance–distance relationships', log values were used because they were more important for the scale classifications. Although we ended up not using those scale variables, we did not waste time redoing all the analyses with untransformed abundance and distance values because the Rs values would have been unchanged. As the reviewer rightly says, the transformation makes no difference to the Spearman's rank values (the log transformation does not affect the ranks). So, while there is no advantage to the Spearman's rank correlations of transforming the data, there is also no

disadvantage. We have removed this sentence from the manuscript to avoid any confusion.

A minor point in the reporting of the statistics: In line 209 and 220, z statistics were presented, with degrees of freedom. But z is usually reserved for Normal tests, where there are no degrees of freedom. I think the authors may have meant t tests? The degrees of freedom are very large, so the t test should be equivalent to a z test, but it is good to maintain correct terminology.

>>>The Z-scores in question here are Fisher's z, not the Z that is the number of standard deviations from the mean in a normal distribution (which the reviewer appears to be talking about). Fisher's z is a transformation of the correlation coefficient (here R_s , which is the same as Pearson's r calculated on the ranks), to achieve more of a normal distribution of R_s as its value nears its limits. To clarify this, we have now changed all instances of this in the manuscript to 'Fisher's Z'.